# Genomic mechanisms of climate adaptation in polyploid bioenergy switchgrass

John T. Lovell[1,38 ✉], Alice H. MacQueen[2,38], Sujan Mamidi[1,38], Jason Bonnette[2,38], Jerry Jenkins[1,38], Joseph D. Napier[2], Avinash Sreedasyam[1], Adam Healey[1], Adam Session[3,4], Shengqiang Shu[3], Kerrie Barry[3], Stacy Bonos[5], LoriBeth Boston[1], Christopher Daum[3], Shweta Deshpande[3], Aren Ewing[3], Paul P. Grabowski[1], Taslima Haque[2], Melanie Harrison[6], Jiming Jiang[7], Dave Kudrna[8], Anna Lipzen[3], Thomas H. Pendergast IV[9,10,11], Chris Plott[1], Peng Qi[9], Christopher A. Saski[12], Eugene V. Shakirov[2,13], David Sims[1], Manoj Sharma[14], Rita Sharma[15], Ada Stewart[1], Vasanth R. Singan[3], Yuhong Tang[16], Sandra Thibivillier[17], Jenell Webber[1], Xiaoyu Weng[2], Melissa Williams[1], Guohong Albert Wu[3], Yuko Yoshinaga[3], Matthew Zane[3], Li Zhang[2], Jiyi Zhang[16], Kathrine D. Behrman[2], Arvid R. Boe[18], Philip A. Fay[19], Felix B. Fritschi[20], Julie D. Jastrow[21], John Lloyd-Reilley[22], Juan Manuel Martínez-Reyna[23], Roser Matamala[21], Robert B. Mitchell[24], Francis M. Rouquette Jr[25], Pamela Ronald[26,27], Malay Saha[16], Christian M. Tobias[28], Michael Udvardi[16], Rod A. Wing[8], Yanqi Wu[29], Laura E. Bartley[30,31], Michael Casler[32,33], Katrien M. Devos[9,10,11,34], David B. Lowry[7,35], Daniel S. Rokhsar[3,4,36,37], Jane Grimwood[1], Thomas E. Juenger[2 ✉] & Jeremy Schmutz[1,3 ✉]

Long-term climate change and periodic environmental extremes threaten food and fuel security[1] and global crop productivity[2–4]. Although molecular and adaptive breeding strategies can buffer the effects of climatic stress and improve crop resilience[5], these approaches require sufficient knowledge of the genes that underlie productivity and adaptation[6]—knowledge that has been limited to a small number of well-studied model systems. Here we present the assembly and annotation of the large and complex genome of the polyploid bioenergy crop switchgrass (*Panicum virgatum*). Analysis of biomass and survival among 732 resequenced genotypes, which were grown across 10 common gardens that span 1,800 km of latitude, jointly revealed extensive genomic evidence of climate adaptation. Climate–gene–biomass associations were abundant but varied considerably among deeply diverged gene pools. Furthermore, we found that gene flow accelerated climate adaptation during the postglacial colonization of northern habitats through introgression of alleles from a pre-adapted northern gene pool. The polyploid nature of switchgrass also enhanced adaptive potential through the fractionation of gene function, as there was an increased level of heritable genetic diversity on the nondominant subgenome. In addition to investigating patterns of climate adaptation, the genome resources and gene–trait associations developed here provide breeders with the necessary tools to increase switchgrass yield for the sustainable production of bioenergy.

Switchgrass (*P. virgatum*) is both a promising biofuel crop and an important component of the North American tallgrass prairie. Historically, tallgrass prairies were one of the largest temperate biomes on Earth, and they remain important sinks for atmospheric carbon[7,8]. However, most extant natural switchgrass populations are restricted to 'relic' sites, which represent crucial but dwindling genetic resources for the future conservation and breeding of tallgrass prairie.

Biomass production is the principal breeding target for switchgrass as a forage and bioenergy crop[9] and is a strong proxy for seed production and evolutionary fitness[10]. Since the US Department of Energy named switchgrass a model herbaceous biofuel feedstock, biomass yield trials have demonstrated the economic viability of switchgrass bioenergy production, and cultivars have been bred that substantially out-produce maize and other cellulosic feedstocks[11]. However, individual cultivars tend to be productive across only a narrow climatic niche. Therefore, to maximize gains, switchgrass breeding and biotechnology should focus on developing climate–genotype matches[12,13] through the identification of the genomic basis of biomass accumulation and climate adaptation in breeding panels. This will bolster future yields[14] and cement switchgrass as an economically and environmentally sustainable bioenergy product.

## The tetraploid switchgrass genome

Although abundant quantitative genetic variation underlies climate-associated stress tolerance and biomass production[15,16], the

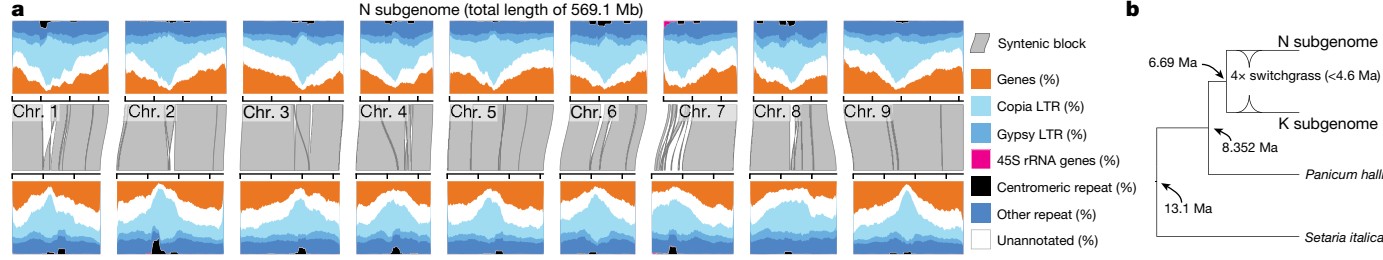

**Fig. 1 | The structure and evolution of the subgenomes of tetraploid switchgrass. a**, Grey polygons (representing $n = 53$ syntenic blocks) demonstrate nearly complete co-linearity between subgenomes. Gene-rich chromosome arms and highly repetitive pericentromeres are typical of grass genomes. LTR, long-terminal repeat. **b**, Subgenome divergence of <4.6 Ma was estimated from a time-scaled phylogenetic tree calibrated to the *Panicum*–*Setaria* node at 13.1 Ma.

fragmented and incomplete nature of previous switchgrass genome sequences have impeded the discovery of candidate genes and other molecular breeding efforts. The genome of the AP13 switchgrass genotype is large (haploid genome size = 1,129.9 megabases (Mb)), repetitive (56.9% repeats) (Fig. 1a, Extended Data Fig. 1) and polyploid. In contrast to some other outcrossing species such as maize (which is represented by the inbred B73 reference genome), AP13 is outbred. Its genome retains a commensurate level of heterozygosity within the range of naturally outcrossing populations (Extended Data Fig. 1). Despite this complexity, our deep PacBio long-read sequencing coupled with deep short-read polishing and bacterial artificial chromosome (BAC) clone validation produced a highly contiguous 'v5' AP13 genome assembly (Extended Data Fig. 1; data are available from Phytozome at https://phytozome-next.jgi.doe.gov). We pruned the resulting large contigs ($N_{50}$ = 5.5 Mb) to a single representative haplotype, and then oriented and ordered into chromosome pseudomolecules using the consensus of two high-density genetic maps (Supplementary Data 1). Chromosomes were assigned to subgenomes via genetic distance to *Panicum rudgeii*[17] (the sister taxa to the K subgenome of *P. virgatum*), and via de novo repeat clustering. The final assembly contains only 0.4% gaps, a 75-fold decrease relative to a previous v4 release from 2016 (https://phytozome-next.jgi.doe.gov/info/Pvirgatum_v4_1). Importantly, the genome assembly was co-linear with three sources of genetic information, despite being assembled independently from all three: the assembly of a close diploid relative (*Panicum hallii*), the marker order of a pseudo-F$_2$ genetic map and the gene order of the alternative subgenome (Fig. 1a, Extended Data Fig. 1, Supplementary Data 2). These co-linearities demonstrated that we have developed a single haploid assembly and annotation for each subgenome.

Crucially, we were able to distinguish gene and repeat sequences between the two subgenomes. The gene annotation—which is derived from Illumina RNA sequencing ($n_{libraries}$ = 88, $n_{conditions}$ = 18, >3 billion reads) and PacBio Iso-Seq ($n_{conditions}$ = 9, > 4.5 million reads, Supplementary Data 3)—encompasses 80,278 primary and 49,664 alternative transcripts and is as complete as the genome assembly (BUSCO = 99.4%) (Extended Data Fig. 1). We leveraged these annotations to build multiple sequence alignments and time-scaled phylogenetic trees, which date subgenome–progenitor species divergence to about 6.7 million years ago (Ma). Long-terminal repeat sequence analysis of subgenome-specific proliferation of retrotransposons sets an upper bound of the polyploidy event that formed switchgrass at ≤4.6 Ma (Fig. 1b), which indicates that tetraploid switchgrass arose during the Pliocene, or the glacial–interglacial cycles of the early Pleistocene epoch.

## Climate adaptation drives biomass yield

Although there are two reproductively isolated[18] switchgrass cytotypes (tetraploid (4×) and octoploid (8×)), tetraploids represent the majority of cultivars[19] and span a broader geographical range than octoploids[20].

To investigate the genetic basis of climate adaptation, stress tolerance and biomass production, we therefore developed a diversity panel of 732 exclusively tetraploid genotypes (Supplementary Data 4). We clonally propagated and transplanted this panel in up to 10 common gardens that spanned 1,862 km of latitude, from southern Texas to South Dakota (USA) ($n_{plants}$ = 5,521) (Fig. 2a) and resequenced each genotype via deep (median = 59×) coverage 2 × 150-bp paired-end PCR-free Illumina libraries. Importantly, resequencing coverage was not biased towards either subgenome (likelihood ratio test $\chi^2$ = 1.32, degrees of freedom = 1, $P$ = 0.25). Our resequencing yielded 33.8 million single-nucleotide polymorphisms (SNPs) (minor allele frequency ≥ 0.5%) mapped against the genome. We also de novo-assembled a 252-genotype subset of these deeply resequenced libraries and called presence–absence and structural variants (for example, 100–1,500-bp insertions and deletions) on the resulting contigs. To connect trait and molecular variation with climate, we extracted 46 climate variables[21,22] from the georeferenced collection location of each genotype and clustered these data into seven groups that explained the majority of climatic variation across the diversity panel (Extended Data Fig. 2).

Climate-associated adaptation in switchgrass has previously been hypothesized to underscore divergence between northern upland and southern lowland ecotypes and is exemplified by divergent leaf and whole-plant morphologies[13,23–26]. In silico classification from morphological data, coupled with ecotype assignments by experts across our diversity panel (Supplementary Data 5), revealed upland ($n = 268$), lowland ($n = 99$) and a third, coastal ecotype ($n = 184$). The coastal ecotype was broadly sympatric with the lowland ecotype but displayed upland leaf characters and lowland plant architecture (Fig. 2a, Extended Data Fig. 2).

We observed strong evidence that adaptive evolution has contributed to ecotype divergence. Whereas winter-kill mortality was rare among northern upland plants (2.4%), nearly half of all coastal (42.1%) and lowland (42.8%) genotypes perished during the winter of 2018–2019 across the 4 northernmost gardens (Fig. 2b). Winter kill was especially severe in the three northwestern plains sites, probably owing to a period of severe cold from late January to early March 2019 (Extended Data Fig. 2). In total, genotypes from the northern 30% of the panel were 218× (Fisher's test odds ratio = 218.17, $P < 1 \times 10^{-15}$) more likely to survive the winter of 2018–2019 in the northern 4 sites than the southernmost 30% of the genotypes.

The latitude gradient across our common gardens also served as the major axis of biomass variation. Among the seven groups of correlated climatic variables, the strongest predictors of biomass variation were always related to temperature (Extended Data Fig. 2). We observed particularly strong signals of extreme 30-year-minimum temperature as a predictor of biomass in the winter-kill-susceptible lowland and coastal ecotypes (Fig. 2c). For both ecotypes, genotypes collected from sites with colder historical extreme minimum temperatures out-performed genotypes from sites with a milder climate in the northern gardens.

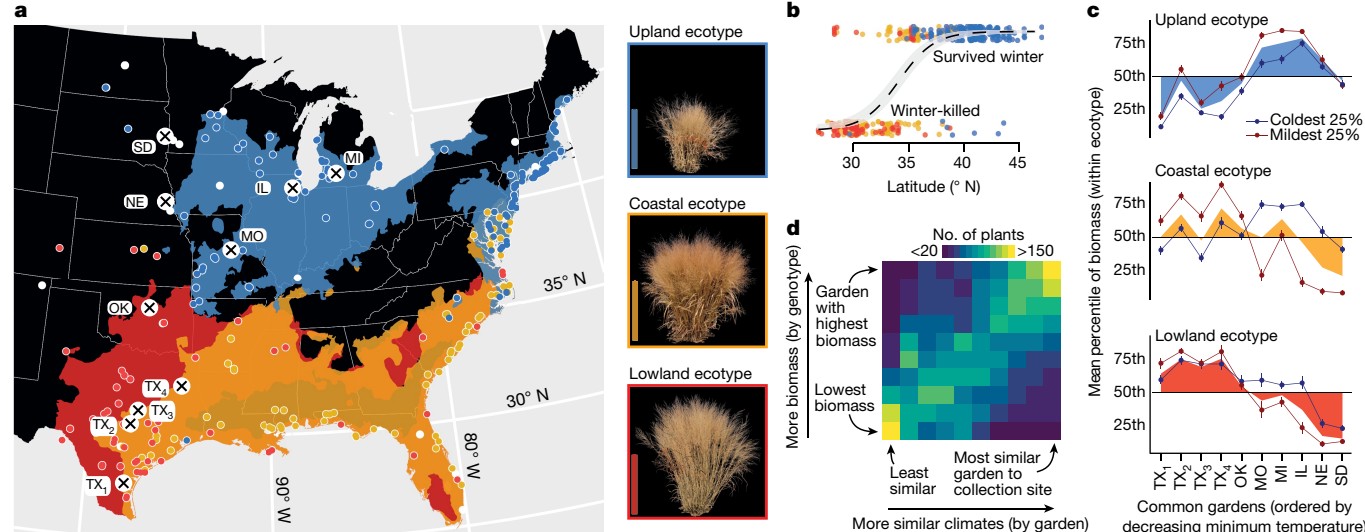

**Fig. 2 | Climatic adaptation within and among switchgrass ecotypes.**
**a**, Geographical distribution of common gardens ($n = 10$) and plant collection locations ($n = 700$ georeferenced genotypes), and spatial distribution models of each ecotype. The ecotype colour legend accompanies the representative images of each ecotype to the right of the map (images were taken at the end of the 2019 growing season and the background was removed with ImageJ (https://imagej.nih.gov/ij)). White-outlined points (coloured by ecotype, or in white if no ecotype assignment was made) indicate the georeferenced collection sites of the diversity panel. The labelled white circles with black crosses indicate the locations of the 10 experimental gardens. Publicly available cultural and physical geographical information system (GIS) layers were accessed with the rnaturalearthdata R package[51]. Scale bars, 1 m. **b**, Across the landscape, survival ($n_{genotypes} = 367$) and winter kill ($n = 184$) in the northern gardens ($n = 3$) was geographically structured: the latitude of the origin of collection site was predictive of survival. A logistic regression prediction (±s.e.) accompanies binary survival along the latitude predictor. **c**, Imputed survival-corrected biomass was converted to percentiles for each ecotype (0 = lowest biomass, 100 = highest) and mean percentiles were plotted overall (coloured polygons, $n = 447$) for each ecotype ($n_{upland} = 211$, $n_{coastal} = 144$, $n_{lowland} = 92$) and garden. The biomass percentiles (mean ± s.e.m.) for the 25% of genotypes from sites with the coldest extreme 30-year coldest minimum temperature (blue lines and points) ($n_{upland} = 52$, $n_{coastal} = 35$, $n_{lowland} = 22$) and the mildest 25% (red lines and points) ($n_{upland} = 53$, $n_{coastal} = 36$, $n_{lowland} = 23$) demonstrate that climate of origin affects biomass within ecotypes and across gardens. **d**, A heat map of the rank of climate similarity ($x$ axis) and imputed biomass ($y$ axis) demonstrates that the majority of 571 genotypes achieve their highest biomass at common gardens that were climatically similar to their source habitat.

However, no climate-of-origin-dependent trade-off was observed in the winter-kill-tolerant upland ecotype. It is possible that a more intensely cold winter than that of 2018–2019 could introduce differential survival in the upland genotypes and produce a trade-off similar to that observed within the two more southern ecotypes. These results add support to our observation that susceptibility to cold temperatures acts both as an agent of natural selection and as a limiter of northern range expansion.

Furthermore, biomass yield for each genotype was generally maximized in the gardens with climates that were most similar to their collection locations (Fig. 2d). As such, local adaptation is manifest not only through survival and stress tolerance, but also through higher biomass accumulation in climates similar to those in which each genotype evolved.

## Ecotype convergence among gene pools

Knowledge of the structure and diversity of gene pools within switchgrass is critical to projecting future gains from molecular breeding and understanding the genetic basis of climate adaptation[12,13]. Several previous population genetic studies of switchgrass assumed that there should be strong correspondence between population genetic structure and the morphological clustering that is used to define ecotypes[20,27,28]. Analysis of our 33.8-million genome-wide SNP database revealed that our diversity panel is strongly subdivided into three major genetic subpopulations that are, in general, geographically distinct (which we refer to as Midwest, Atlantic and Gulf) (total $F_{ST} = 0.27$) (Fig. 3a). The clustering of presence–absence and structural variants largely recapitulates SNP-based subpopulation structure (Extended Data Fig. 3), providing consistent evidence of subpopulation differentiation that may include large-effect mutations at several molecular scales.

Population genetic structure was discordant with variation in morphological ecotype, which segregated strongly within genetic subpopulations. Plants with upland ecotype traits were present in both the Atlantic (37%) and Midwest (63%) gene pools. Similarly, 54% and 46% of coastal ecotype accessions were assigned to Atlantic and Gulf subpopulations, respectively (Fig. 3a). All plants with lowland morphology were clustered within the Gulf subpopulation. However, these Gulf lowland plants had approximately equal proportions of individuals that survived and perished during the northern winter (Fig. 2c). Thus, important genetic diversity for breeding was present within genetic subpopulations—a pattern that was validated through realized genetic gains of biomass and winter survival within several switchgrass breeding populations[29,30].

Despite ecotypic convergence among subpopulations, coalescent simulations dated the divergence of the subpopulations to the mid-Pleistocene epoch (>358,000 generations (0.7–1.4 Ma, assuming a 2–4 year generation time)) (Extended Data Fig. 3). Thus, extant switchgrass gene pools have been diverging for nearly half of the evolutionary history of polyploid switchgrass. In contrast to the deep sequence divergence among subpopulations, we observed very little molecular genetic differentiation between upland and coastal ecotypes within the Atlantic subpopulation ($F_{ST} = 0.03$), or between lowland and coastal ecotypes within the Gulf subpopulation ($F_{ST} = 0.03$) (Extended Data Fig. 3).

Admixture appears to be common between the Gulf and Atlantic subpopulations; comparisons of plants with coastal ecotype traits from both of these subpopulations were molecularly more similar ($F_{ST} = 0.19$) than for noncoastal Gulf and Atlantic plants ($F_{ST} = 0.24$). By contrast, the plants with upland morphologies in the Midwest and Atlantic subpopulations were no more similar than other plants from those subpopulations ($F_{ST}$ for both = 0.30). This convergence of upland morphologies in two highly differentiated genetic subpopulations

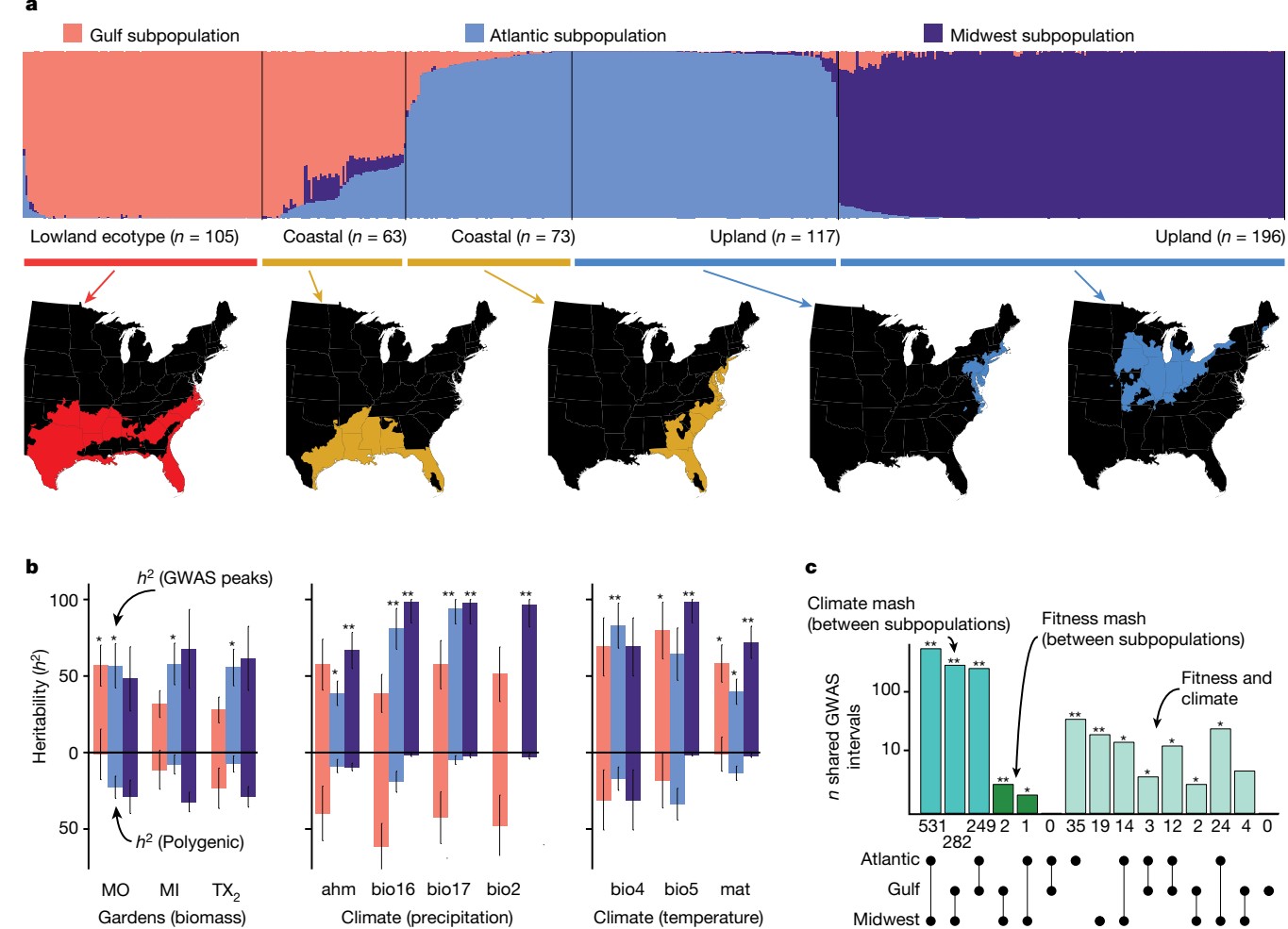

**Fig. 3 | Population and quantitative genomics of climate-associated adaptation. a**, Admixture proportions among three gene pools (coloured by subpopulation) and three ecotypes (labelled below), calculated using eigenvector decomposition of the identity-by-descent matrix. The corresponding geographical distribution of each ecotype is presented below the bar plot (coloured by the ecotype distributions from Fig. 1a). Publicly available cultural and physical GIS layers were accessed with the rnaturalearthdata R package[51]. **b**, Post hoc tests of SNP–heritability (mean $h^2 \pm$ s.e.m.) attributable to polygenic background (below the black horizontal lines) and significant multivariate adaptive shrinkage GWAS hits (above the black horizontal lines) are presented for the three main sites (biomass) and for precipitation- and temperature-related climate variables, and coloured by subpopulations (following **a**). Extended Data Fig. 2c provides descriptions of the climate variables (ahm, bio2, bio4, bio5, bio16, bio17 and mat). Statistical significance of higher heritability for GWAS hits relative to polygenic inheritance is indicated for two-sided $Z$-score $P$ values; **$P < 0.001$, *$P < 0.05$. **c**, There are large and significant overlaps in climate-associated multivariate adaptive shrinkage (mash) intervals between subpopulations, and smaller but significant overlaps between fitness and climate hits in the Atlantic and Midwest subpopulations. Two-sided Fisher's test $P$ value significance, following **b**.

could be the result of independent genetic origins of the upland ecotype or rare but evolutionarily important[31] admixture events. We evaluate these hypotheses below.

## Genetic targets for yield improvement

To detect the genetic basis of climate adaptation and fitness within the diversity panel, we conducted multivariate adaptive shrinkage[32] on genome-wide association mapping (GWAS) results within and across genetic subpopulations. Multivariate adaptive shrinkage shares GWAS peak effect size and direction between univariate tests to improve power to detect significant, shared results. Multivariate adaptive shrinkage results were determined for both fitness GWAS (which mapped winter survival and biomass in the three largest common gardens (MI, MO and TX₂)), and climate GWAS (which detected associations between SNP variation and the climate of origin (seven representative climate variables)). To make direct comparisons among subpopulations (which

have different segregating SNPs), we summarized the 12,239 significant linkage-disequilibrium block 'peaks' of multivariate adaptive shrinkage ($\log_{10}$-transformed Bayes factor > 2)[33] into 10,090 20-kb regions (20 kb represents the inflection point at which linkage disequilibrium decay flattens) (Extended Data Fig. 3) for climate ($n_{regions} = 9,856$) and fitness ($n_{regions} = 332$) GWAS (Supplementary Data 6). A weighted list of candidate genes—including putative SNP effects, the existence of presence–absence or structural variants, gene co-expression and physical proximity to the GWAS peaks—can be found in Supplementary Data 7.

GWAS peaks explained the majority of heritable phenotypic and climatic variation (SNP–heritability) both across and within gene pools (Fig. 3b). SNP–heritability of fitness ($h^2 = 51.5 \pm 15.4\%$ (mean $\pm$ s.e.m.)) and climate-associated peaks ($h^2 = 70.5 \pm 14.0\%$) collectively explained over threefold-more variation than the polygenic background (fitness = $19.5 \pm 9.1\%$, climate = $18.2 \pm 9.5\%$) (Extended Data Table 1). The high heritability of these climate and biomass associations indicated that relatedness at a small subset of all variants out-predicted overall

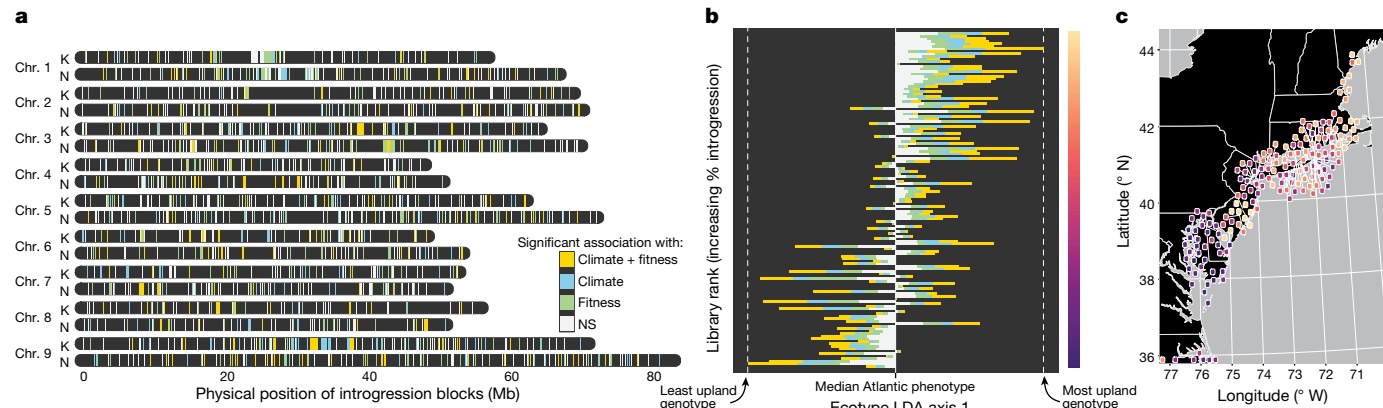

**Fig. 4 | Mapping the location and effect of Midwest introgressions in the Atlantic subpopulation. a**, Positions of all high-frequency (present in >10 genotypes, $n = 1,640$) introgressions from Midwest into the Atlantic subpopulation are coloured by significance in the two redundancy analyses for climate (blue, $n = 234$), biomass and survival (green, $n = 329$) or 'climate–fitness overlap', which are significant in both (gold, $n = 245$). NS, not significant. **b**, Introgressions are strongly associated with a more upland phenotype among 135 genotypes. For each genotype, the position along the first discriminant axis between ecotypes (Extended Data Fig. 2) is scaled relative to the median Atlantic ecotype value, then plotted and coloured by the proportion of introgressed sequence in each significance bin from **a**. **c**, The introgression ranks from **b** were converted to a purple–orange colour scale (right of **b**) and georeferenced positions of collection sites for each library are plotted for the northern Atlantic seaboard of the USA. Publicly available cultural and physical GIS layers were accessed with the rnaturalearthdata R package[51].

relatedness and provides breeders with genetic diversity to target for switchgrass improvement in local environments.

Loci that are associated with both fitness and climate of origin are probably involved in local adaptation[34], and are strong targets for the breeding of locally adapted cultivars. Overall, we observed nearly 2× more overlap of 20-kb regions associated with both climate and fitness than expected by chance (Fisher's test odds ratio = 1.92, $P < 1 \times 10^{-6}$). This overlap was especially strong within the two northern subpopulations (Midwest, odds = 11.5× and $P < 1 \times 10^{-15}$; Atlantic, odds = 17.8× and $P < 1 \times 10^{-15}$) (Fig. 3c), where we expected to see the strongest effect of selection on survival during cold winters.

Many regions of climate and fitness overlap were polymorphic only within a single genetic subpopulation, which highlights several, possibly independent, genetic paths to climate adaptation in switchgrass. However, 9.5% (940) of the 20-kb climate intervals were polymorphic in several genetic subpopulations. Given the substantial evidence of admixture between the Gulf and Atlantic subpopulations (Fig. 3a), we expected that contemporary gene flow would be the major contributor to shared polymorphisms. Contrary to this hypothesis, the majority (511 regions) of all multi-subpopulation GWAS intervals were shared between the two most genetically distinct gene pools (Atlantic and Midwest). Given the deep divergence time between these subpopulations, rare or ancient gene flow[35] may have created these shared adaptive polymorphic regions.

## Evolutionary convergence via introgression

To explicitly address how introgressions may have shaped the distribution of climate–SNP associations, we investigated physically contiguous regions of admixture across the genome using a hidden Markov model[36]. Introgressions between subpopulations represented 2.98% of the content of our resequenced genomes (Fig. 4a), but were >1.5× more likely to contain shared GWAS intervals across subpopulations than expected by chance (Fisher's test odds ratio = 1.55, $P < 1 \times 10^{-8}$), indicating that adaptive introgressions underlie at least a portion of heritable variants shared among subpopulations.

Of particular interest were a suite of introgressions from the Midwest to the Atlantic subpopulation that dated to about 8,700 generations before present (17–34 thousand years ago (ka)), which coincides with a northern range expansion after the Last Glacial Maximum (about 22 ka).

Atlantic genotypes with higher levels of Midwest introgressions exhibited a more-upland suite of traits (Fig. 4b) and were overrepresented along the northern margin of the otherwise subtropical and temperate range of the Atlantic subpopulation (Fig. 4c). Consistent with adaptive roles for genomic introgressions in other systems[31,37], these findings suggest that introgression of putatively northern-adapted alleles from the Midwest into the Atlantic subpopulation could have facilitated the post-glacial colonization by switchgrass of colder habitats in the northeastern coastal region of the USA. To test this hypothesis, we conducted redundancy analyses to relate the presence of introgression blocks with climatic, geographical and phenotypic factors. Overall, Midwest introgressions in the Atlantic subpopulation were over four times more strongly associated with climate (percentage of variance explained = 46.5%) than geography (11.5%). Although 532 and 651 introgressions from the Midwest to the Atlantic subpopulation were associated with climate of origin or biomass, respectively, 254 introgressions were outliers for both analyses—representing a nearly 7-fold enrichment over expectations of independence between each set (odds ratio = 6.99, $P < 1 \times 10^{-15}$). These results reinforce the hypothesis that Midwest introgressions have shaped the climatic niche and phenotypic distribution of the northern Atlantic genotypes and support a growing body of evidence that demonstrates that adaptive introgressions can facilitate both range expansion and ecotype evolution[38,39].

## Reduced heritability of dominant subgenomes

Polyploidy is common among lineages of flowering plants and can increase the genetic diversity available to selection[40,41], which can lead to adaptive evolution or sorting that alters ecological niche characteristics[42]. This process may explain the generally greater prevalence of polyploids in poleward latitudes and higher elevations that were once covered by ice sheets during glacial cycles[43].

Genes duplicated during the formation of a polyploid can subfunctionalize (divide ancestral gene functions among paralogous genes), neofunctionalize (evolve new gene function for paralogues) or simply be lost[44]. Following polyploid speciation, one subgenome commonly retains more genes and exhibits, on average, higher expression levels than the other subgenome, a phenomenon known as subgenome dominance[45]. As with other polyploids[46–48], subgenome dominance and subfunctionalization were clear in switchgrass. Relative to the

N subgenome, the K subgenome had higher gene density (77.4 versus 68.0 genes per Mb, binomial $P < 1 \times 10^{-15}$), more upregulated genes (5,445 versus 4,477, binomial $P < 1 \times 10^{-15}$) and lower rates of mutation accumulation (5,255 genes in the K subgenome with a synonymous mutation rate ($K_s$) greater than that in the N subgenome, versus 6,751 genes in the N subgenome with $K_s$ greater than that in the K subgenome, binomial $P < 1 \times 10^{-15}$). Combined, all 11 of our subgenome statistics (Extended Data Fig. 4) point to stronger evolutionary constraint of and bias towards the K subgenome, which suggests that the potential for adaptive evolution may be differentially partitioned between subgenomes.

Given the evolutionary biases towards retention of the K subgenome, we expected to see stronger signals of climate adaptation[44], biomass and survival among SNPs on the K subgenome. Instead, 75.9% of biomass SNP–heritability was attributable to the N subgenome, and only 24.1% to the K subgenome across the 10 common gardens (Extended Data Fig. 4). Furthermore, 54.3% of Midwest introgressions into the Atlantic subpopulation were found on the N subgenome, a significant enrichment (binomial test $P < 1 \times 10^{-7}$), even when correcting for the 7.5% expansion of the N subgenome (binomial test $P = 0.0012$). The abundance of introgressions and heritable biomass variation attributable to the N subgenome may appear to be at odds with subgenome evolutionary biases towards the K subgenome. One potential explanation for this counterintuitive finding is that relaxed evolutionary constraint (reduced purifying selection) on the N subgenome may have allowed for accumulation of adaptive genetic variation through directional or diversifying selection. As such, the N subgenome has accumulated heritable variation[49] that future breeding regimes can target to shape natural switchgrass populations and improve biofuel yield.

## Discussion

As the climate and the natural environment change, it is increasingly critical to qualify expectations of genetic improvements in domesticated species and the adaptive potential of wild populations[50]. Indeed, plant genomes offer glimpses into the past and future of crop and wild plant populations. Adaptation to glacial–interglacial cycles offers an instructive analogue for current and future environmental change, one that we explore here to investigate the past, present and future genomic mechanisms of climate adaptation and yield improvement in switchgrass.

However, the complexity of plant genomes has also presented a major barrier to the development of genetic resources that facilitate fast and effective molecular breeding. Our methodology and success in sequencing the complex genome of switchgrass will facilitate ecological and agricultural genomics in nearly any system. For example, our results demonstrate that adaptation to northern climates has been facilitated by introgressions between anciently diverged subpopulations, which provides further support for the hypothesis that admixture between divergent genomes can enhance adaptation to novel environments[37]. Such adaptive introgressions and heritable subgenome-specific genetic variation[49] may provide the genetic paths of least resistance that permit colonization of novel habitats during periods of environmental variability. Combined, obligate outcrossing and polyploidy—traits that are often consciously avoided when selecting genomic study systems—are the primary drivers of switchgrass adaptation in nature and the sources of genetic variation available for selection to improve biofuel yield through a changing future.

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

[1]Genome Sequencing Center, HudsonAlpha Institute for Biotechnology, Huntsville, AL, USA. [2]Department of Integrative Biology, University of Texas at Austin, Austin, TX, USA. [3]Department of Energy Joint Genome Institute, Lawrence Berkeley National Laboratory, Berkeley, CA, USA. [4]Department of Molecular and Cell Biology, University of California, Berkeley, Berkeley, CA, USA. [5]Department of Plant Biology, Rutgers University, New Brunswick, NJ, USA. [6]Plant Genetic Resources Conservation Unit, USDA-ARS, Griffin, GA, USA. [7]Department of Plant Biology, Michigan State University, East Lansing, MI, USA. [8]Arizona Genomics Institute, University of Arizona, Tucson, AZ, USA. [9]Institute of Plant Breeding, Genetics and Genomics, University of Georgia, Athens, GA, USA. [10]Department of Crop and Soil Sciences, University of Georgia, Athens, GA, USA. [11]Department of Plant Biology, University of Georgia, Athens, GA, USA. [12]Department of Plant and Environmental Sciences, Clemson University, Clemson, SC, USA. [13]Department of Biological Sciences, Marshall University, Huntington, WV, USA. [14]School of Biotechnology, Jawaharlal Nehru University, New Delhi, India. [15]School of Computational and Integrative Sciences, Jawaharlal Nehru University, New Delhi, India. [16]Noble Research Institute LLC, Ardmore, OK, USA. [17]Department of Agronomy and Horticulture, University of Nebraska, Lincoln, NE, USA. [18]Department of Agronomy, Horticulture and Plant Science, South Dakota State University, Brookings, SD, USA. [19]Grassland, Soil and Water Research Laboratory, USDA-ARS, Temple, TX, USA. [20]Division of Plant Sciences, University of Missouri, Columbia, MO, USA. [21]Environmental Science Division, Argonne National Laboratory, Lemont, IL, USA. [22]Kika de la Garza Plant Materials Center, USDA-NRCS, Kingsville, TX, USA. [23]Plant Breeding Department, Antonio Narro Agrarian Autonomous University, Saltillo, Mexico. [24]Wheat, Sorghum, and Forage Research Unit, USDA-ARS, Lincoln, NE, USA. [25]Texas A&M AgriLife Research and Extension Center, Texas A&M University, Overton, TX, USA. [26]Department of Plant Pathology and the Genome Center, University of California, Davis, Davis, CA, USA. [27]Joint BioEnergy Institute, Emeryville, CA, USA. [28]Western Regional Research Center, USDA-ARS, Albany, CA, USA. [29]Department of Plant and Soil Sciences, Oklahoma State University, Stillwater, OK, USA. [30]Department of Microbiology and Plant Biology, University of Oklahoma, Norman, OK, USA. [31]Institute of Biological Chemistry, Washington State University, Pullman, WA, USA. [32]US Dairy Forage Research Center, USDA-ARS, Madison, WI, USA. [33]DOE Great Lakes Bioenergy Research Center, University of Wisconsin, Madison, WI, USA. [34]DOE Center for Bioenergy Innovation, Oak Ridge, TN, USA. [35]DOE Great Lakes Bioenergy Research Center, Michigan State University, East Lansing, MI, USA. [36]Center for Advanced Bioenergy and Bioproducts Innovation, Berkeley, CA, USA. [37]Chan-Zuckerberg Biohub, San Francisco, CA, USA. [38]These authors contributed equally: John T. Lovell, Alice H. MacQueen, Sujan Mamidi, Jason Bonnette, Jerry Jenkins. ✉e-mail: jlovell@hudsonalpha.org; tjuenger@mail.utexas.edu; jschmutz@hudsonalpha.org

# Methods

No statistical methods were used to predetermine sample size. The experiments were completely randomized, and investigators were not aware of genotype identifiers while conducting experiments or sequencing.

## Plant collections, propagation, cultivation and phenotyping

To form the diversity panel, seeds, rhizomes and clonal propagules from natural and common garden sources were collected from 2010 to 2018. Plants grown from seed followed a standard growth procedure[16]. In brief, 10–15 seeds were sown in 9-cm square pots containing a mixture of ProMix BX potting soil (Premier Tech Horticulture) and Turface MVP calcined clay (Turface Athletics) and vernalized for 7 days at 4 °C. Pots were then placed in a lit greenhouse with 14-h day length and 30-°C/22-°C day/night temperature. Seedlings were thinned at the 3-leaf stage to 1 plant per pot and allowed to grow until the 5-tiller stage. Rhizome propagules and 5-tiller seedlings were transferred to 5-gallon pots containing finely ground pine bark mulch (Lone Star Mulch) and time-release fertilizer (Osmocote 14-14-14, ScottsMiracleGro). All individual plants were propagated in Austin by clonal division from 2016 to 2018, targeting >10 clones per unique accession. Cleary 3336F systemic fungicide (Cleary Chemicals) was applied to the plants as necessary to control fungal pathogens. Plants were placed in 1-gallon pots for the final propagation.

Planting in the field sites occurred from 15 May to 10 July 2018 and followed previously published methods[16]. In brief, plants were transported to each site by truck, where each field was covered with one layer of DeWitt weed cloth. Plants were placed in holes that were cut into the weed cloth into a honeycomb design in which each plant had four nearest neighbours, all located 1.56 m from one another. To prevent edge effects, the lowland Blackwell cultivar was planted at every edge position. Plants were hand-watered following transplantation. Aboveground portions of all plants were left to stand over the winter of 2018–2019 and removed in the spring of 2019 before spring tiller emergence. At the end of the 2019 season, plants were tied upright as a bunch and harvested with sickle bar mowers.

We generated two measures of fitness for the 2019 growing season: log-transformed biomass (kg) and proportion of winter survival (Supplementary Data 8). Biomass data were obtained from all living individuals during harvest in October and November 2019. Plants with an estimated mass <750 g were placed in paper bags and dried whole at 60 °C until no additional moisture loss occurred, then weighed for total dry biomass. Plants with an estimated mass >750 g were weighed in the field for wet biomass on a hanging scale with a ±5-g resolution. To determine biomass of these plants, approximately 500 g of whole tillers were subsampled from each plant, weighed, dried as above and reweighed. The wet biomass of the whole-plant sample was then multiplied by the per cent moisture in the subsample to approximate total dry biomass. Plants were considered to have experienced winter mortality during the 2018–2019 winter season when no new growth was seen from plant crowns by 1 June 2019. The dead plant crowns were excised from the experiment and replaced with plants of the Blackwell cultivar in July or September 2019.

## Genome assembly and polishing

We sequenced the Alamo switchgrass genotype AP13 using a whole genome shotgun sequencing strategy and standard sequencing protocols at the Department of Energy Joint Genome Institute and the HudsonAlpha Institute for Biotechnology. The genome was assembled and polished from 4,520,785 PacBio reads (121.66× raw sequence coverage from a total of 59 P6C4 2.0 and 2.1 chemistry cells with 10-h movie times and a p-read yield of 91.76 Gb) (Extended Data Fig. 1) using the MECAT assembler[52] and ARROW polisher[53]. Final genome polishing and error correction was conducted with one 400 bp insert 2 × 150 bp Illumina HiSeq fragment library (177.1×). Reads with >95% simple sequence repeats and reads <50 bp after trimming for adaptor and quality ($q < 20$, 5-bp window average) were removed. The final read set consisted of 1,259,053,614 reads for a total of 168× coverage of high-quality Illumina bases. This produced an initial diploid assembly of 6,600 scaffolds (6,600 contigs), with a contig $N_{50}$ of 1.1 Mb, 3,489 scaffolds larger than 100 kb and a total 2C (diploid) genome size of 2,013.4 Mb.

Assembling a haploid genome in an outbred individual, such as AP13, will generally yield both haploid copies in heterozygous regions, necessitating computational steps to represent each chromosome as a single-copy haplotype without duplicate copies being unnecessarily repeated. Our initial assembly was approximately double the expected haploid (1C) genome size of 1.2 Gb. Therefore, to detect putative meiotically homologous haplotypes, we identified and counted shared 24-mers that occurred exactly twice in the assembly and binned contigs accordingly. A total of 3,152 shorter and redundant alternative haplotypes and 2,387 overlapping contig ends were identified, comprising a total sequence of 871.2 Mb. The remaining 1,142.2 Mb of sequence was ordered and oriented into 18 chromosomes by aligning genetic markers from 2 available maps (Supplementary Data 1) to the MECAT assembly; 563 joins and 57 breaks were made, with 10,000 Ns representing the unsized gap sequence. Overall, 97.2% of the assembled sequence was contained in the chromosomes. Telomeric sequence was identified using the $(TTTAGGG)_n$ repeat and properly oriented. The remaining scaffolds were screened against GenBank bacterial proteins and organelle sequences and removed if found to match these sequences. To resolve minor overlapping regions on contig ends, adjacent contig ends were aligned to one another using BLAT[54]; a total of 47 adjacent duplicate contig pairs were collapsed.

We conducted two rounds of error correction. First, we corrected homozygous SNPs and insertions and/or deletions (indels) by aligning the Illumina 2 × 150 bp library to the release consensus sequence using bwa mem[55] and identifying homozygous SNPs and indels with the UnifiedGenotyper tool of GATK[56]. A total of 690 homozygous SNPs and 80,199 homozygous indels were corrected in the release. Second, we computationally finished 11,343 assembled contigs sequenced from BAC clones with a combination of ABI 3730XL capillary sequencers[57] and single index Illumina clone pools and aligned this set of switchgrass clones to the SNP-fixed genome to find heterozygous SNPs that were out of phase with their neighbours. To resolve these phase-switched alleles, the full set of the raw PacBio reads was aligned to the assembly. For each read, the phase of each heterozygous site was determined and 62,732 out-of-phase heterozygous sites were corrected.

To distinguish the N and K subgenomes, we used a de novo repeat-clustering method and validated this with phylogenetic distances to a related species. We searched for 'diagnostic' 15-mers via Jellyfish[58] in LTR regions of Gypsy, Copia and Pao insertions (identified by RepeatMasker[59] and LTRHarvest[60]) that distinguished each set of homologous chromosomes (≤1 hit in one homologue and ≥100 in the other). The LTR sequences that shared common 15-mers were grouped as superfamilies and were aligned within each superfamily by BLAST. Superfamily members with significant BLAST hits ($e < 0.01$, ≥90% length) were assigned into families and aligned by Mafft[61]. Jukes–Cantor distances between LTR families were computed by the R ape package[62], and clustered into two distinct sets of subgenomes. Clustering was identical between LTRs and alignments to *P. rudgei* (K.M.D. and E. Kellogg, unpublished data), which is an ancient relative of the K subgenome[17], giving high confidence that we have effectively assigned all chromosomes to the correct subgenomes. Finally, we assigned chromosome identifiers and oriented each chromosome pseudomolecule via synteny with *Setaria italica*[63]. The final haploid version 5.0 release contained 1,125.2 Mb of sequence, consisting of 626 contigs with a contig N50 of 5.5 Mb and a total of 97.2% of assembled bases in chromosomes.

## Gene annotation

Transcript assemblies were made from about 2 billion pairs of 2 × 150-bp stranded paired-end Illumina RNA-seq reads, about 1 billion pairs of 2 × 100-bp paired-end Illumina RNA-seq reads and 454 reads (Supplementary Data 3) using PERTRAN (details of which have previously been published[64]). In brief, PERTRAN conducts genome-guided transcriptome short-read assembly via GSNAP[65] and builds splice alignment graphs after alignment validation, realignment and correction. In total, around 4.5 million PacBio Iso-Seq circular consensus sequences[66] were corrected and collapsed, resulting in approximately 677,000 putative full-length transcript assemblies. Subsequently, 668,176 transcript assemblies were constructed using PASA[67] from RNA-seq reads, full-length cDNA, Sanger expressed sequence tags, and corrected and collapsed PacBio circular consensus sequence reads. Loci were determined by EXONERATE[68] alignments of switchgrass transcript assemblies and proteins from *Arabidopsis thaliana*[69], soybean[70], Kitaake rice[71], *Setaria viridis*[72], *P. hallii* var. *hallii*[64], *Sorghum bicolor*[73], *Brachypodium distachyon*[74], grape and Swiss-Prot[75] proteomes. These alignments were accomplished against a repeat-soft-masked switchgrass genome using RepeatMasker[59] (repeat library from RepeatModeler[76] and RepBase[77]) with up to 2,000-bp extension on both ends unless extending into another locus on the same strand. Incomplete gene models, which had low homology support without full transcriptome support, or short single exon genes (<300-bp coding DNA sequences (CDS)) without protein domain or good expression were removed.

## Comparative genomics

Syntenic orthologues and paralogues were inferred for the two switchgrass subgenomes via the GENESPACE pipeline[64], using default parameters and two outgroups: *P. hallii* var. *hallii*[64] and *S. bicolor*[73]. In brief, GENESPACE parses protein similarity scores into syntenic blocks and runs orthofinder[78] on synteny-constrained blast results. The resulting block coordinates and syntenic orthology networks give high-confidence anchors for evolutionary inference.

To calculate the ancestral states of CDS regions, we first determined sequences that share common ancestry using genomes from Phytozome[79]. The final number of hits to the switchgrass genome were 38,960 and 33,772 for *P. hallii*, and *S. bicolor*, respectively. For any given orthology network, we built two multiple sequence alignments in mafft[61], one excluding the focal switchgrass sequence ($msa_0$) and one forcing $msa_0$ to align to the coordinate system of the focal sequence via the --keeplength parameter. We then extracted marginal character states with the maximum likelihood algorithm in Phangorn[80]. For each reconstruction, only the internal node closest to the switchgrass branch was used as the ancestral state. Overall, we analysed 40,943 switchgrass gene models (216,157 exons) covering 54.95 Mb (Supplementary Data 9).

## Subgenome evolution and dating

To infer the ages of the subgenomes and tetraploid switchgrass, we took a conservative set of orthologues with simple 2:1:1 networks between *P. virgatum*, *P. hallii* and *S. italica*. This yielded 45,045 switchgrass proteins aligning to 24,549 *P. hallii* proteins, resulting in 20,496 homologue pairs and 4,053 singletons (2,396 for K subgenome and 1,660 for N subgenome) from the cross-species analysis. We aligned the translated CDS of these sequences using Dialign-TX[81]. The aligned CDS sequences were concatenated and fed to Gblocks[82] using default parameters. Gblocks filtered the alignment of 18,044,244 CDS nucleotides to 16,321,302 positions, in 50,334 blocks. The resulting alignment was then used in PhyML[83] to build a maximum-likelihood tree using the general-time reversible model. This tree was used as an input to r8s[84], to compute a time tree and calibrate the *Panicum–Setaria* node of the tree to 13.1 Ma[63]. To date subgenome divergence and therefore the timing of polyploid switchgrass speciation, we leveraged burst distances, which refer to all distances within an LTR family (whereas pairwise distances refer to the distance between the

5′ and 3′ LTRs of the same insertion). The 5′ versus 3′ distances of the N- or K-subgenome-specific retrotransposons were used to date the insertion times of those elements. This method cannot be used for the *P. virgatum*-specific or *Panicum*-specific families because the more recent expansions of those elements dominate the distributions. Instead, we relied on comparing the best cross-species alignments to estimate the LTR distances of the *P. virgatum–P. hallii* and *Panicum–Setaria* nodes. This way, we have calibration points to compare the LTR distances to the more confident protein-coding gene divergences between species.

## Subfunctionalization and gene expression analyses

To assess whether the subgenome evolution biases observed at the protein-coding sequence scale were manifest in phenotypes, we explored gene expression biases between homologues from biologically replicated AP13 leaf tissue ($n \geq 5$) collected at two sites (TX$_2$ and MI). Illumina paired-end RNA-seq 150-bp reads were quality trimmed ($Q \geq 25$) and reads shorter than 50 bp after trimming were discarded. High-quality sequences were aligned to *P. virgatum* v5.1 reference genome using GSNAP[65] and counts of reads uniquely mapping to annotated genes were obtained using HTSeq v.0.11.2[85]. The test for differential expression was conducted through a likelihood ratio test in DESeq2[86]. Library sizes were calculated before splitting the reads by subgenome; these sizes were used as the size factors in the analysis of differential expression. Subfunctionalization was defined as a significant subgenome-by-environment interaction from the likelihood ratio test. Subgenome expression bias was tested for both the field gardens and annotation libraries using post hoc Wald-test contrasts between subgenomes within conditions. Significant bias was defined as differential expression false-discovery-rate-adjusted $P < 0.05$. Weighted gene coexpression clustering of AP13 gene annotation RNA-seq libraries was conducted with WGCNA[87] with a power of 6. Raw counts can be found in Supplementary Data 10.

## Ploidy assessment

We used a LSRFortessa SORP Flow Cytometer (BD Biosciences) to determine ploidy levels of the resequenced accessions. For each plant, 200–300 mg of young leaf tissue was macerated in a Petri dish with a razor blade and treated for 15 min with 1 ml Cystain PI Absolute P nuclei extraction buffer (Sysmex Flow Cytometry) mixed with 1 µl 2-mercaptoethanol. Samples were filtered to isolate free nuclei with a CellTrics 30-µm filter (Sysmex) and treated for 20 min on wet ice with 2 ml of Cystain PI Absolute P staining buffer (Sysmex), 12 µl of propidium iodide and 6 µl of RNase A. Samples were run on the flow cytometer to determine nuclei size with a minimum of 10,000 nuclei analysed per sample. Output from the flow cytometer was analysed with FlowJo software (BD Biosciences) and samples were binned into three categories on the basis of the average units of fluorescence per nuclei (Supplementary Fig. 1). Ploidy level of the sample was considered 4× if the cell population had 40,000–80,000 units of fluorescence, 6× for 80,000–100,000 units and 8× for 100,000–140,000 units. The binning parameters were established with flow cytometry data from several *P. virgatum* accessions of known ploidy.

We also assessed ploidy of the samples via the distribution of variant allele frequency at biallelic SNPs (as described in 'Variant calling'). This method assumes that tetraploids and octoploids follow different allele frequency distribution patterns, with tetraploids having 0.5/0.5 (reference and variant depths) and octoploids having a mixture of 0.75/0.25 and 0.5/0.5. If the proportion of hits with $0.48 \leq x \leq 0.52$ was <0.035, the library was considered octoploid and if it was ≥0.035, tetraploid; 837 out of 870 samples (96.2%) that had flow cytometry data matched with these results.

## Variant calling

A total of 789 tetraploid diversity samples were resequenced at a median depth of 59× (range 20×–140×). Of these, 732 were used for further

analysis after filtering for missing data, outlier elevated heterozygosity and collection site discrepancies. The samples were sequenced using Illumina HiSeq X10 and Illumina NovaSeq 6000 paired-end sequencing (2 × 150 bp) at HudsonAlpha Institute for Biotechnology and the Joint Genome Institute. To account for different library sizes, reads were pruned to ≤50× coverage, then mapped to the v5 assembly using bwa-mem[55].

SNPs were called by aligning Illumina reads to the AP13 reference with BWA-mem. The resulting .bam file was filtered for duplicates using Picard (http://broadinstitute.github.io/picard) and realigned around indels using GATK 3.0[56]. Multi-sample SNP calling was done using SAMtools mpileup[88] and Varscan V2.4.0[89] with a minimum coverage of eight and a minimum alternate allele count of four. Genotypes were called via a binomial test. SNPs within 25 bp of a 24-mer repeat were removed from further analyses. Only SNPs with ≤20% missing data and minor allele frequencies >0.005 were retained, resulting in 33,905,042 SNPs across 75% of the genome at a coverage depth between 8× and 500×. Phasing was performed using SHAPEIT3[90]. $F_{ST}$ calculations were accomplished via vcftools[91]. We tested for subgenome read-mapping bias by generating mean coverage per Mb for each of the 732 libraries and 18 chromosomes. We then fit a mixed effects linear model to these data in lme4[92] in which the chromosome number (1–9) was a random effect, to test the main effect of subgenome. Models with and without the main effect term were compared via a likelihood ratio test.

Individual de novo assemblies for the 732 short read libraries were constructed using HipMer[93] with a $k$-mer size of 101 to maximize haplotype splitting among contigs. As the assemblies varied in quality and contiguity, the sample set considered for gene presence–absence and structural variant detection was narrowed to 251 samples (pan-genome set) based on total assembly size, contig $N_{50}$ length and total gene alignments per library.

To assess presence–absence variation of genes across the pan-genome, we aligned all AP13 proteins and a unique set of 6,161 proteins from *Oropetium thomaeum* ($n_{proteins} = 1,476$)[94], *S. italica* ($n = 1,085$)[63], *Setaria viridis* ($n = 891$)[72], *P. hallii* var. *filipes* ($n = 1,048$)[64], *S. bicolor* ($n = 878$)[95] and *P. hallii* var. *hallii* ($n = 772$)[64]. These unique genes were extracted from single-copy orthology networks inferred via orthofinder[78] and selection owing to a lack of orthology to switchgrass. All proteins (≥100 amino acids) were aligned to all de novo assemblies using BLAT[54]. Gene alignments from AP13 proteins were considered present if they aligned with greater than or equal to 80% identity and 75% coverage, whereas other grass proteins were considered present with alignments greater than 70% identity and 75% coverage (to allow greater divergence among species). Variable (pan-genome shell) genes (considered present across 40–60% of the population; $n = 5,432$) were extracted from the presence–absence variation matrix and used to visualize differences among non-admixed individuals from the Atlantic, Gulf and Midwest subpopulations. Testing genes that were significantly over- or under-represented within each subpopulation was conducted with a $\chi^2$ test with a Benjamini–Hochberg multiple testing correction ($P \leq 0.05$).

To detect structural variants across the pan-genome, contigs (≥2 kb) from each library were aligned to the AP13 reference genome using ngmlr[96] with default settings for PacBio reads. The resulting .bam file was sorted using samtools[88] and used for calling structural variants with sniffles[96]. Individual structural variant calls were merged across samples using SURVIVOR[97], with a maximum allowed distance of 1 kb. The resulting .vcf file was filtered using bcftools[88] using a minimum minor allele frequency of 0.1, and considering only insertions and deletions between 100 and 1,500 bp in length.

## Population genomics

To assess the genetic population structure of the 732 tetraploid libraries (Supplementary Data 4), we extracted all fourfold degenerate sites (putatively neutral) with ancestral state calls (Supplementary Data 9) from the ancestral state alignments. This list of sites, which represents our highest confidence neutral loci, was then linkage-disequilibrium-pruned using a threshold of $|r| \leq 0.6$, resulting in 59,789 sites for downstream analyses in the R package SNPRelate[98].

The extent of linkage disequilibrium for the population was determined from SNPs[99] in PLINK[100]. Linkage disequilibrium ($r^2$) was calculated using plink (--ld-window 500 --ld-window-kb 2000). The $r^2$ value was averaged every 500 bp. A nonlinear model was fit for this data in R using the nls function, and the extent was determined as to when the linkage disequilibrium ($r^2$) nonlinear curve stabilized.

Population genetic structure was assessed hierarchically. Given the presence of highly divergent ecotypes across the study range, we first analysed the broadest genetic population structure using discriminant analysis of principal components (DAPC)[101] in adegenet v.2.0.1[102]. This method does not rely on common assumptions (for example, Hardy–Weinberg equilibrium and linkage disequilibrium) that underlie many population clustering approaches and therefore provides a valuable tool to look at broad structural divisions. DAPC demonstrated a strong set of gene pools and separated Midwest genotypes from all others. We then evaluated the genetic population structure and potential admixture of the remaining non-Midwest individuals using a Bayesian clustering algorithm implemented in STRUCTURE v.2.3.4[103] via the admixture model with correlated allele frequencies. The analysis consisted of 20,000 burn-in steps and 30,000 replicates of 1–6 genotypic groups, each of which was run 10 times. Ancestry coefficients across all subpopulations were assigned post hoc through eigenvector decomposition in SNPRelate.

We inferred the demographic history of the switchgrass samples using Multiple Sequentially Markovian Coalescent (MSMCv.2.0[104]), which is a population genetic method used to infer demographic history and population structure through time from sequence data. This method models an approximate version of the coalescent under recombination, and produces tests of both population size and divergence time. MSMC was run using four haplotypes for each subpopulation, skipping ambiguous sites, an estimated rhoOverMu of 0.25 and a time segment pattern of $10 \times 2 + 20 \times 5 + 10 \times 2$. We estimated rhoOverMu as 0.25 as the mean value from 100 iterations without the fixed recombination parameter for 5 sets of 4 haplotypes in each subpopulation and averaged them. To estimate scaled divergence time in generations, we assumed a mutation rate of $6.5 \times 10^{-8}$. To make estimates of initial divergence time, we compared adjacent relative cross-coalescence rate (RCCR) values (past to present) (Supplementary Data 11). If there was a decline, either at a single time segment or within contiguous segments or within two interleaved time segments (>0.01; observed range 0.01–0.28), and the following neighbours were nearly zero (≤0.009; observed range: −0.1–0.009), we considered that to be a starting point for population separation. However, if there was another decline within five time segments, we considered the latter as the start of population separation. We replicated the analyses with 16 sets of different individuals for each subpopulation contrast.

Population structure was visualized across SNPs, structural variants and presence–absence variants via eigenvector decomposition of a distance matrix. First, a Euclidean distance matrix was calculated among 0/1/2 (reference homozygote, heterozygous, alternative homozygote) library × marker matrices for each of the three variant call types. The Euclidean matrix was then scaled and centred to remove among-library coverage variance via Gower's centred similarity matrix, implemented in the R package MDMR[105].

## Ecotype classification

Mature switchgrass accessions at or near anthesis were surveyed for 16 plant traits (leaf: length, width, length/width ratio, area, lamina thickness and lamina/midrib thickness ratio; whole plant: number of tillers, tiller height, product of tiller height × number, tiller height/count ratio, panicle height, panicle height/count ratio, leaf canopy height and tiller/

leaf height ratio; phenology: date of green-up and date of panicle emergence) to determine ecotype identity during the summer of 2019 at the University of Texas J. J. Pickle Research Campus (PKLE; or $TX_2$ (Austin, Texas, USA) and Michigan State University Kellogg Biological Station (KBSM; or MI (Hickory Corners, Michigan, USA)) common gardens (see Supplementary Data 5 for detailed descriptions of these variables). The phenology measurements, including green-up (when the first green vegetative structures emerge from the rhizome crown) and panicle emergence (when the first reproductive structures emerge from the tiller), were assayed daily. Detailed leaf morphology was assessed on a representative leaf of each plant by measuring length and width (in mm), midrib and lamina thickness (in μm) (Mitutoyo 547-500S caliper) and leaf area (in $mm^2$) (Licor 3100C leaf area meter). In addition to these quantitative traits, we also generated a qualitative upland–lowland index for both the leaf and whole-plant appearance, collected at the end of the summer 2019 in Austin ($TX_2$ site). Each plant characteristic was assessed on a 1–5 scale from most lowland-like to most upland-like. The established cultivars Alamo and Dacotah were used for baseline measurements of lowland and upland characters, respectively. Plant characters assessed included: tiller appearance, from thickest and most lowland-like to thinnest and most upland-like; leaf appearance, from widest, longest and most lowland-like to shortest, thinnest and most upland-like; canopy colour from bluest and most typically lowland to darkest green and most typically upland. This visual approach is akin to basic selection criteria often used by switchgrass breeders.

To assess phenotypic structure in these data, we used a DAPC[101]. Prior groups were determined by first transforming the phenotypic data using principal component analysis (PCA), then the first 10 principal components were used in a $k$-means algorithm to classify individuals into 3 possible groupings aiming to maximize the variation between groups. Next, DAPC was implemented on the 10 retained principal components to provide an efficient description of the ecotypic clusters using two synthetic variables, which are linear combinations of the original phenotypic variables that have the largest between-group variance and the smallest within-group variance (that is, the discriminant functions).

We classified each of the 651 tetraploid genotypes surveyed for the 16 traits at the MI and $TX_2$ gardens (34 total features, 32 quantitative and 2 qualitative ordinal traits) to 1 of the 3 ecotypes through a low-capacity neural network with 1 hidden layer and 5 units (Supplementary Data 5). The neural network was implemented in caret[106] and was trained on seven cultivars with known ecotypes (lowland: Kanlow and Alamo; coastal: High Tide and Stuart; upland: Summer, Dacotah and Sunburst) and 78 additional genotypes that were in the same SNP-based genetic cluster (Extended Data Fig. 3), collected in the same states and clustered most closely in phenotypic PCA space with the exemplar cultivars. These high-affinity exemplar genotypes are printed in Supplementary Data 5. Ecotypes for the remaining 582 genotypes that were phenotyped for the ecotype classification traits were predicted with caret[106]. By using traits collected at gardens representing both the northern and southern switchgrass range, we hoped to avoid local climate bias on plant phenotype and subsequent ecotype classification. Furthermore, the neural network classification approach offers one notable advantage over both DAPC and expert's qualification: because the neural network is anchored to known and published genotypes, experimentation that includes these common cultivars will be able to more effectively recapitulate our assignments.

## Admixture and introgression block calculation and dating

We built a database of admixture-informative SNPs through a two-step pipeline. First, ancestry coefficients were calculated as in 'Population Genomics' from fourfold degenerate sites that had associated ancestral-state calls. The 30 samples with the least missing data and proportion of genome-wide admixture ≤ 0.001 for each subpopulation were used to define subpopulation-specific allele frequencies. These

libraries were used to find SNPs with at least one pairwise $F_{ST}$ value >0.4, as calculated with the 'W&C84' method in the snpRelate function snpGdsFst. Second, these global ancestry-informative sites were parsed within each subpopulation to those with minor allele frequencies > 0.05 and missingness < 0.05. These sites were further pruned within subpopulations first to sites with $|r| < 0.9$ (10 SNPs or 1,000-bp windows), then to $|r| < 0.95$ (1,000 SNPs or 10,000-bp windows) in snpRelate. This process resulted in the following SNP and library counts for each subpopulation: Atlantic, 579,468 SNPs and 284 libraries; GULF, 641,975 SNPs and 215 libraries; and Midwest, 481,563 SNPs and 196 libraries.

To test for the physical locations of admixture blocks between each pair of subpopulations, we used Ancestry_HMM[36,107]. This approach leverages allele frequencies in putative parental populations to determine regions of likely introgressions in a test population. For each of the three subpopulations, we sought to determine the timing, extent and current positions of admixture block introgressions. In each case, we permitted two pulses from each of the other two subpopulations. Ancestry_HMM can optimize the number of generations before present when an ancestry pulse occurred and the proportion of individuals involved in the admixture pulse. However, 8-parameter optimization with >480,000 sites and >150 libraries was not computationally feasible. Therefore, we optimized parameters using 40 randomly sampled libraries with admixture coefficients within the 0.2–0.8 quantiles of the admixture proportion distribution and SNPs only on chromosome 4 of the N subgenome. We chose this chromosome as representative of others because of a lack of obvious large high-frequency introgressions. The resulting ancestry pulse parameter optimizations were founded on an initially unadmixed population 10,000 generations before present, and two subsequent admixture pulses for each of the other two subpopulations; the optimized pulses are as follows (source–reference): Midwest–Atlantic ($n_{generations} = 8,658$ and $P_{admixed} = 0.001\%$; 67 and 0.7%), Gulf–Atlantic (85 and 1.1%; 17 and 0.25%), Atlantic–Gulf (79 and 1.9%; 11 and 0.38%), Midwest–Gulf (79 and 0.86%; 11 and 0.14%), Atlantic–Midwest (66 and 0.27%; 14 and 0.036%), and Gulf–Midwest (71 and 0.15%; 14 and 0.033%). These pulses were supplied to the full model with all individuals and chromosomes, along with an error probability of 0.001, maximum number of generations before present of 10,000 and effective population size of 100,000. Posterior ancestry probabilities were decoded into haplotype blocks and blocks were binned into clusters of similarly positioned blocks.

## Landscape genomics

Geographical maps were made with publicly available layers downloaded from Natural Earth (https://www.naturalearthdata.com/). Various plotting routines rely on the sf[108] and raster[109] packages in the R environment for statistical computing[110]. Climate data were downloaded from WorldClim[22] (19 bioclimatic variables, 0.5-arcmin resolution 1960–2000) and ClimateNA[21]. The distribution of climate variables across collections sites was explored via dynamic clustering[111] followed by partitioning around medoids clustering[112] with $k = 7$. The most representative climate variables were defined as those most correlated with the first eigenvector of variation within each cluster. Six of the seven clusters included WorldClim variables.

Weather data were downloaded from the NOAA portal for the most proximate weather station to each garden site that had complete daily temperature (minimum–maximum), and precipitation data from 1 September 2018 to 31 October 2019. The NOAA weather station identifiers used for each garden are as follows: IL (USC00110338), MI (USW00014815), MO (USW00003945), NE (USC00255362), OK (USW00053926), SD (USC00391076), $TX_1$ (USC00414810), $TX_2$ (USC00410433), $TX_3$ (USC00418862) and $TX_4$ (USW00003901).

Climate–phenotype associations across gardens were conducted on both raw data and imputed data. Latitude–survival associations (Fig. 2b) were accomplished on raw data with logistic regressions via glm with a binomial family in R. Imputations, which were accomplished

in base R using nearest neighbours across all available phenotypes ($k = 5$), were used exclusively for tests of the rank order of gardens (Fig. 2c, d). Climate similarity–biomass associations were accomplished in mixed linear models via lmer[92], comparing the full model (fixed = climate distance + intercept, random = genotype identifier) to a reduced model without the climate distance fixed effect using a likelihood ratio test.

Species distribution modelling (SDM) was used to simulate modern-day potential ranges for all ecotypes (upland, lowland and coastal) of *P. virgatum*. The final datasets used to build the SDMs comprised 277 (upland), 199 (coastal) and 121 (lowland) occurrence records. Six environmental predictors were used in our final SDM modelling (BIO1 = annual mean temperature, BIO2 = mean diurnal range, BIO4 = temperature seasonality, BIO5 = maximum temperature of warmest month, BIO16 = precipitation of wettest quarter and BIO17 = precipitation of driest quarter). SDMs were then generated with BIOMOD2 v.3.3[113] with seven modelling algorithms: generalized linear models, boosted regression trees, artificial neural networks, flexible discriminant analysis, random forest, classification tree analysis and multivariate adaptive regression splines. For each model, the occurrence data were coupled with 500 pseudo-absence data generated randomly within the modelled study area with equal weighting for presences and pseudo-absences[114]. Models were trained with 80% of the coupled occurrences and pseudo-absence data and tested with the remaining 20%. Each modelling algorithm was run 100 times for a total of 700 models, which were evaluated via true skill statistics (TSS)[115]. TSS values ranging from 0.2 to 0.5 were considered poor, from 0.6 to 0.8 useful, and >0.8 good to excellent[116]. Unique ensemble SDMs were computed from approximately the 50 best SDMs out of 700 models for the three ecotypes on the basis of TSS threshold values (upland TSS threshold = 0.96, lowland TSS threshold = 0.93 and coastal TSS threshold = 0.965). The final ensemble SDMs were projected onto present climate layers to visualize modern-day potential ranges (Supplementary Data 12).

We examined how the presence of Midwest introgressions in the Atlantic subpopulation were associated with the independent and joint influences of climate, geography and kinship, by implementing redundancy analysis in vegan[117–120]. To partition explainable variance in introgression presence attributable to climate, kinship and geography, we ran four models: one full model with introgression presence (potential introgression blocks were coded as 0 for Atlantic inheritance or 1 for Midwest introgression) explained by climate (that is, the seven representative climate variables), kinship (the first two principal components calculated from the set of putatively neutral markers) and geography (latitude and longitude), and three models for each of these three factors conditioned on the other two. The inertia (that is, variance) values from the constrained matrix of each model were compared to determine the relative importance of climate, kinship, geography and their joint effect. Furthermore, to find introgression regions strongly linked to climate and survival-corrected biomass, we extracted the loadings for the redundancy analysis axes from two additional models: (1) one predicted by only climate and (2) one predicted only by survival-corrected biomass. Both models were significant according to permutation tests ($n = 999$; $P < 0.001$ for both), and all axes were approximately normally distributed. SNPs loading at the tails of each axis were more likely to indicate selection related to the predictors (that is, climate or survival-corrected biomass), so we identified all markers that were at least 2.5 s.d. (two-tailed $P = 0.012$) from the centre as introgressions putatively under selection[119].

## GWAS

Owing to the large sizes of our common garden datasets, we developed a pipeline—the switchgrassGWAS R package (https://github.com/Alice-MacQueen/switchgrassGWAS)—to allow fast, less-memory-intensive GWAS on the diversity panel, and to analyse

the extent to which SNP effects were similar or different for phenotypes measured at different sites. This package leverages bigsnpr[121] to perform fast (>300× faster than TASSEL) statistical analysis of massive SNP arrays encoded as matrices. It also incorporates current gold standards in the human genetics literature for SNP quality control, pruning and imputation, as well as population structure correction in GWAS. To test the significance of many effects in many conditions (for example, multiple sites, climate variables and so on), we used mashr[32], a flexible, data-driven method that shares information on patterns of effect size and sign in any dataset for which effects can be estimated on a condition-by-condition basis for many conditions and SNPs. We determined which SNPs had evidence of significant phenotypic effects using local false sign rates, which are analogous to false discovery rates but more conservative (in that they also reflect the uncertainty in the estimation of the sign of the effect)[122]. We used these values to find SNPs with $\log_{10}$-transformed Bayes factors > 2. Here, the Bayes factor was the ratio of the likelihood of one or more significant phenotypic effects at a SNP to the likelihood that the SNP had only null effects. Following previous work[33], a Bayes factor of $>10^2$ is considered decisive evidence in favour of the hypothesis that a SNP has one or more significant phenotypic effects.

To calculate regional heritability for climate- and fitness-associated SNPs we followed a previously described two-step method[123]. Variance component analysis was accomplished with ASReml (VSN International), using genomic relationship matrices calculated using the van Raden method[124]. Genomic relationship matrices were calculated within each subpopulation and for the full diversity panel. A kinship matrix based on all SNPs used in the univariate GWAS was calculated ($G$), as well as a kinship matrix based on SNPs significantly associated with climate in that subpopulation ($\log_{10}$-transformed Bayes factor > 2; $Q_{climate}$) and a kinship matrix based on SNPs significantly associated with biomass or winter survival in that subpopulation ($\log_{10}$-transformed Bayes factor > 2, or >1.385 for Gulf subpopulation; $Q_{fitness}$). These kinship matrices were used for regional heritability mapping[123] as in a previous publication[125], using mixed models of the form:

$$\mathbf{y} = 1 + Zu + Zv + e$$

$$\mathrm{Var}(u) = G\sigma_u^2$$

$$\mathrm{Var}(v) = Q\sigma_v^2$$

$$\mathrm{Var}(e) = I\sigma_e^2$$

in which the vector $\mathbf{y}$ represents the biomass values, $Z$ is the design matrix for random effects, $u$ is the whole genomic additive genetic effect, $v$ is the regional genomic additive genetic effect and $e$ is the residual. Matrix $G$ is the whole genomic relationship matrix using all SNPs for the whole genome additive effect. Matrix $Q$ is the regional genomic relationship obtained as above: one of $Q_{climate}$ or $Q_{fitness}$. $I$ is the rank-$y$ identity matrix, in which $y$ is equal to the number of biomass values. Whole genomic, regional genomic and residual variances are $\sigma_u^2$, $\sigma_v^2$ and $\sigma_e^2$, respectively. Phenotypic variance ($\sigma_p^2$) is $\sigma_u^2 + \sigma_v^2 + \sigma_e^2$. Whole genomic heritability, regional heritability and total heritability are $h_u^2 = (\sigma_u^2/\sigma_p^2)$, $h_v^2 = (\sigma_v^2/\sigma_p^2)$ and $h_{u+v}^2 = (\sigma_u^2 + \sigma_v^2/\sigma_p^2)$, respectively.

These models were run for the three locations where subpopulation GWAS were conducted: Columbia, Missouri; Hickory Corners, Michigan; and Austin, Texas. This resulted in 80 models: 4 sets of populations (the full diversity panel and 3 subpopulations), 2 model types (one model with $G$ only and a $G + Q$ model), for 10 phenotypes (biomass at 3 sites and 7 environmental variables).

Variance component analyses were also used to partition variance between the K- and N-subgenomes. Only SNPs with ancestral state calls (Supplementary Data 9) were used in this analysis, resulting in 460,429

SNPs used for each population subset. Kinship matrices based on all SNPs on a particular chromosome were calculated ($Q_{Chr01K}$ to $Q_{Chr09K}$, and $Q_{Chr01N}$ to $Q_{Chr09N}$), resulting in 18 kinship matrices. These kinship matrices were used for regional heritability mapping, using mixed models of the form:

$$\mathbf{y} = 1 + Zv_{1K} + Zv_{1N} + Zv_{2K} + \ldots + Zv_{9N} + e$$

$$\mathrm{Var}(v_i) = Q_i \sigma_{v_i}^2$$

$$\mathrm{Var}(e) = I \sigma_e^2$$

in which the vector $\mathbf{y}$ represents the biomass values, $Z$ is the design matrix for random effects, $v_{1K}$ (to $v_{9K}$) or $v_{1N}$ (to $v_{9N}$) (collectively designated $v_i$) are the chromosome-specific genomic additive genetic effects and $e$ is the residual. Matrices $Q_i$ are the chromosome-specific genomic relationship matrices for the nine chromosomes of the N and K subgenomes. Chromosome-specific and residual variances are $\sigma_{v_i}^2$ and $\sigma_e^2$, respectively. Chromosome-specific heritability is $h_{v_i}^2 = (\sigma_{v_i}^2 / \sigma_p^2)$, and subgenome-specific heritability is the sum of these variances across the nine chromosomes within each subgenome.

## Candidate gene exploration

We integrated multiple data structures to rank and provide meaningful culling criteria for candidate genes within introgression intervals and physical proximity to quantitative trait loci peaks. In the case of GWAS peaks, candidate genes were defined as those loci within a 20-kb interval surrounding the mashr peak. Candidate genes for genomic introgressions must have at least partially overlapped the introgression interval. As inference of GWAS and introgressions were conducted within genetic subpopulations, all statistics reported in Supplementary Data 7 (candidate gene lists) are also subpopulation-specific, with the exception of gene co-expression analysis (which was conducted only on AP13 RNA-sequencing libraries used for annotation purposes (Supplementary Data 3)). For a given interval, we present a set of statistics. First, the physical proximity to the peak location was calculated as the midpoint of the gene to the midpoint of the interval (introgression) or GWAS peak position. Second, as the causal locus underlying GWAS peaks within a subpopulation must necessarily be variable within that subpopulation, we extracted all SNPs within and proximate to candidate gene models. These variants were annotated with SNPeff[126] and the weighted sum of three main categories of variants (high, moderate and low; a description of these can be found at https://pcingola.github.io/SnpEff/se_inputoutput/#effect-prediction-details) for each gene were calculated as SNPeff_score = high × 20 + moderate × 5 + low × 1. Third, for each gene, we calculated the minor allele frequency of structural and presence–absence variants. Fourth, we include a vector of the identity of the WGCNA clusters for each gene. Finally, if the candidate was a homologue of flowering-time GWAS candidate genes from a previous publication[127], the identity of the overlapping interval or gene is included.

## Reporting summary

Further information on research design is available in the Nature Research Reporting Summary linked to this paper.

## Data availability

Sequence Read Archive accession codes for all RNA and DNA sequencing libraries can be found in Supplementary Data 3 and 4, respectively. The v5 AP13 genome has been deposited at DDBJ/ENA/GenBank under the accession JABWAI000000000. The genome, gene and repeat annotations can also be downloaded directly from Phytozome at https://phytozome-next.jgi.doe.gov/info/Pvirgatum_v5_1. Whenever possible, plant material will be shared upon request. Source data are provided with this paper.

## Code availability

Custom pipelines for GWAS and other analyses are available from dataverse at https://doi.org/10.18738/T8/J377KE.

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

**Acknowledgements** Plant collecting was conducted in collaboration with J. Randall (North Carolina Botanical Garden) through the Seeds for Success programme, A. Stottlemeyer (OSU and the USDA-NIFA Biotechnology Risk Assessment Grant Program, no. 2010-33522-21703), T. Quedensley, M. Donahue, D. Schemske and J. M. M. Reyna. We thank the Brackenridge Field laboratory, the Ladybird Johnson Wildflower Center and the Juenger laboratory for support with plant care and propagation. M. Donahue led the curation, propagation and maintenance of the diversity panel. Fieldwork was also conducted by P. Duberney, S. Reeder, K. Turner, M. Carey, T. Arredondo, N. Ryan, B. Watson, B. Battershell, N. Albert, H. Wilson, L. Simon, J. Sanley, L. Vormwald, T. Bortnem, S. Hofmann, M. Iceberg, C. Lamb and T. Vugteveen. Advice from J. G. Monroe, D. Hoover, P. Edger, J. Lasky, E. Kellogg, J. Vogel, G. Sarath and J. Tuskan helped to craft experimental designs, sequencing strategies and earlier versions of this text. R. VanBuren, P. Edger, H. Zheng, D. Ware and L. Cattivelli provided genome comparison information. We thank the HudsonAlpha Genomic Services Lab for loading Illumina X10 sequencing runs. This research was supported by the US Department of Energy Awards DESC0014156 to T.E.J., DE-SC0017883 to D.B.L. and DE-SC0010743 to K.M.D., the Great Lakes Bioenergy Research Center (Awards DESC0018409 and DE-FC02-07ER64494) and the Center for Bioenergy Innovation (Award DE-AC05-000R22725). Funding was provided by National Science Foundation PGRP Awards IOS0922457 and IOS1444533 to T.E.J. and IOS1402393 to J.T.L. The work conducted by the US Department of Energy Joint Genome Institute is supported by the Office of Science of the US Department of Energy under Contract No DE-AC02-05CH11231. The work conducted by the Joint BioEnergy Institute is supported by the Office of Science of the US Department of Energy under contract no. DE-AC02-05CH11231. The work conducted by Argonne National Laboratory is supported by the Office of Science of the US Department of Energy under contract DE-AC02-06CH11357. J.S. thanks T. Marsh for transferring his passion for ecological science to J.S. T.E.J. thanks K. Robertson for introducing him to prairie habitats and plant diversity.

**Author contributions** K.D.B., P.R., M. Saha, L.E.B., M.C., K.M.D., D.B.L., D.S.R., J.G., T.E.J. and J.S. designed research. S.B., M.H., J. Jiang, T.H.P. IV, S.T., J.Z., J.M.M.-R., P.R., C.M.T., M.U., M.C., K.M.D. and D.B.L. contributed plant material and resources. J. Jenkins, C.P. and S.S. assembled and annotated the genome. J.B., A.R.B., P.A.F., F.B.F., J.D.J., D.B.L., J.L.-R., R.M., R.B.M., F.M.R. Jr, M. Saha, Y.W. and T.E.J. designed and executed field experiments. K.B., L.B., C.D., S.D., A.E., D.K., A.L., E.V.S., D.S., M. Sharma, R.S., A. Stewart, V.R.S., Y.T., J.W., X.W., M.W., Y.Y., M.Z. and R.A.W. conducted sequencing and data acquisition. J.T.L., A.H.M., S.M., J.D.N., A. Session, A. Sreedasyam, P.P.G., T.H., A.H., P.Q., C.A.S., G.A.W. and L.Z. conducted statistical and computational analyses. The manuscript was written by J.T.L., A.H.M., T.E.J. and J.S. with contributions from all authors.

**Competing interests** The authors declare no competing interests.

**Additional information**
**Correspondence and requests for materials** should be addressed to J.T.L., T.E.J. or J.S.

**a** *Genome contiguity*

| Genome attribute | Size / Value |
|---|---|
| Scaffold total | 626 |
| Contig total | 1,090 |
| Scaffold sequence total | 1,129.9 Mb |
| Contig sequence total | 1,125.2 Mb (0.4% gap) |
| Contig N50 | 5.5 Mb |
| Chromosome Sequence | 1,093.8 Mb (97.2%) |

**b** *Genome assembly libraries*

| Sequencing Platform | Mean read / Insert Size | n. reads (M) | Assembled Coverage (x) |
|---|---|---|---|
| Illumina | 500 | 1,325.3 | 177.12 |
| PACBIO | 10,758* | 6.983 | 83.42 |
| Total | N/A | 1,332.3 | 260.54 |

*Mean length of PacBio reads (bp)

**d** *Collinearity between physical position and genetic mapping position*

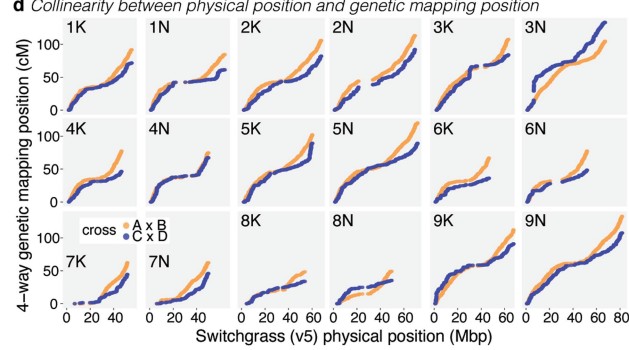

**e** *Genome annotation statistics*

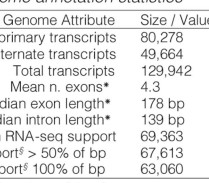

| Genome Attribute | Size / Value |
|---|---|
| n. primary transcripts | 80,278 |
| n. alternate transcripts | 49,664 |
| Total transcripts | 129,942 |
| Mean n. exons* | 4.3 |
| Median exon length* | 178 bp |
| Median intron length* | 139 bp |
| n. with RNA-seq support | 69,363 |
| n. support$^§$ > 50% of bp | 67,613 |
| n. support$^§$ 100% of bp | 63,060 |

*for primary transcripts only
$^§$gene models with 50% (100%) of bases covered by RNA-seq reads

**f** *Heterozygosity among 732 genotypes*

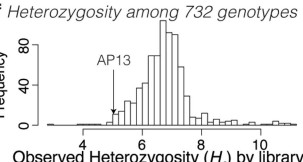

**c** *Genome assembly statistics and metadata for a selection of outbred or polyploid plants*

| Species (version) | Genome Size (Mbp) | Ploidy | Pedigree | Heterozygous bases / Mb | %repeat$^‡$ | Contiguity (N50, kb) |
|---|---|---|---|---|---|---|
| Switchgrass (AP13 v5) | 1,125 | 4x | Outbred | 5,398 | 56.9% | 5,500 |
| Maize (B73 refgen_v4) | 2,106 | 2x* | Inbred | 0$^§$ | 60.2%$^¶$ | 1,280 |
| Durum wheat (cv. Svevo) | 10,450 | 4x | Inbred | 0$^§$ | 82.2% | 56 |
| Broomcorn millet (Pm_0390_v0.1) | 923 | 4x | Inbred | 0$^§$ | 58.2% | 369 |
| Teff (Dabbi v3) | 576 | 4x | Inbred | 1,100 | 26.5% | 1,550 |
| Poplar (P. trichocarpa v4.1) | 422 | 2x* | Outbred | 6,836 | 28.4% | 13,100 |
| Soybean (Wm82.a4 v.1) | 978 | 2x* | Inbred | 59 | 46.7% | 419 |
| Cotton (G. hirsutum v3.1) | 2,305 | 4x | Inbred | 6 | 77.7% | 389 |
| Walnut (Chander v2) | 567 | 2x | Outbred | 3,800 | 58.4% | 1,100 |
| Strawberry (x ananassa) | 813 | 8x | Outbred | ~5,000$^†$ | 36.0% | 79 |

*Diploid species often described as paleopolyploid; $^§$No heterozygosity, genome was assembled as completely inbred
$^†$Heterozygosity was not directly calculated, this is an estimate; $^‡$%of genome in repeatMasker gff3 or reported; $^¶$LTR only

**g** *Synteny between subgenomes and P. hallii*

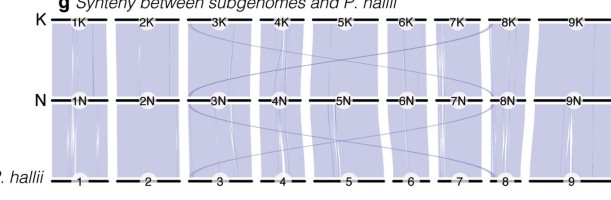

**Extended Data Fig. 1 | Genome assembly and annotation. a–c**, Genome contiguity (**a**) and library coverage (**b**) demonstrate that the v5 release is a very complete genome and that it is among the best available plant reference genomes (**c**), compared to maize[128] durum wheat[129], broomcorn millet[130], teff[131], poplar[132], soybean[70], cotton[47], walnut[133] and strawberry[46]. **d**, Complete collinearity between marker order in both crosses (number of markers = 4,701) of a 4-way mapping population is evident. **e**, Genome annotation statistics present a gene annotation that is as complete as the assembly. **f**, Observed heterozygosity ranges from <4 to >10% among our 732-library resequencing panel. **g**, Nearly the entire single-copy genome of *P. hallii* is syntenic with both switchgrass subgenomes; pale blue polygons represent syntenic blocks between subgenomes and *P. hallii*. The one exception is a previously known over-retained region representing the ρ duplication on Chr. 03 and 08[64].

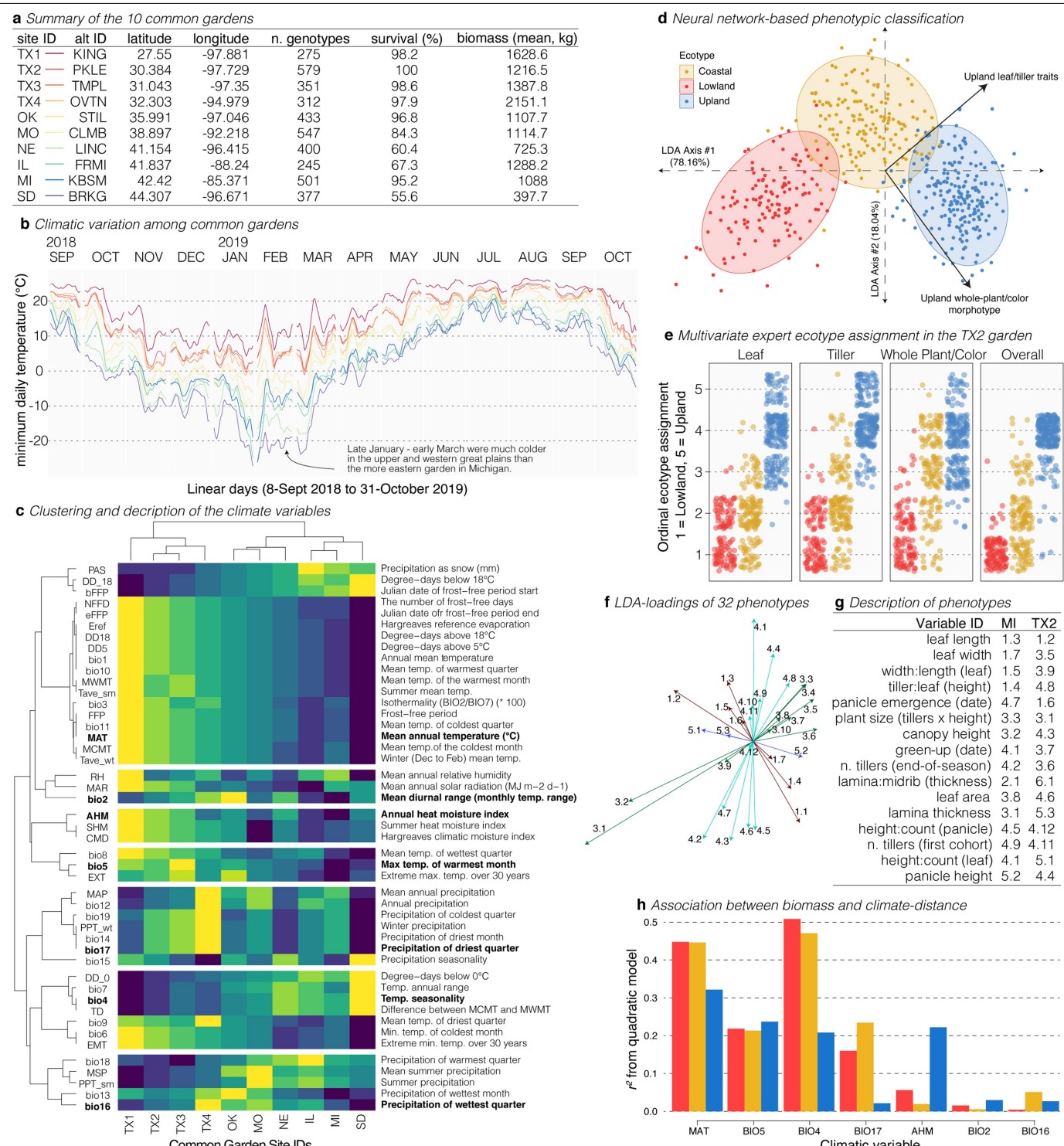

**a** *Summary of the 10 common gardens*

| site ID | alt ID | latitude | longitude | n. genotypes | survival (%) | biomass (mean, kg) |
|---|---|---|---|---|---|---|
| TX1 | KING | 27.55 | -97.881 | 275 | 98.2 | 1628.6 |
| TX2 | PKLE | 30.384 | -97.729 | 579 | 100 | 1216.5 |
| TX3 | TMPL | 31.043 | -97.35 | 351 | 98.6 | 1387.8 |
| TX4 | OVTN | 32.303 | -94.979 | 312 | 97.9 | 2151.1 |
| OK | STIL | 35.991 | -97.046 | 433 | 96.8 | 1107.7 |
| MO | CLMB | 38.897 | -92.218 | 547 | 84.3 | 1114.7 |
| NE | LINC | 41.154 | -96.415 | 400 | 60.4 | 725.3 |
| IL | FRMI | 41.837 | -88.24 | 245 | 67.3 | 1288.2 |
| MI | KBSM | 42.42 | -85.371 | 501 | 95.2 | 1088 |
| SD | BRKG | 44.307 | -96.671 | 377 | 55.6 | 397.7 |

**b** *Climatic variation among common gardens*

Late January - early March were much colder in the upper and western great plains than the more eastern garden in Michigan.

**c** *Clustering and decription of the climate variables*

PAS — Precipitation as snow (mm)
DD_18 — Degree–days below 18°C
bFFP — Julian date of frost–free period start
NFFD — The number of frost–free days
eFFP — Julian date ofr frost–free period end
Eref — Hargreaves reference evaporation
DD18 — Degree–days above 18°C
DD5 — Degree–days above 5°C
bio1 — Annual mean temperature
bio10 — Mean temp. of warmest quarter
MWMT — Mean temp. of the warmest month
Tave_sm — Summer mean temp.
bio3 — Isothermality (BIO2/BIO7) (* 100)
FFP — Frost–free period
bio11 — Mean temp. of coldest quarter
**MAT** — **Mean annual temperature (°C)**
MCMT — Mean temp.of the coldest month
Tave_wt — Winter (Dec to Feb) mean temp.
RH — Mean annual relative humidity
MAR — Mean annual solar radiation (MJ m−2 d−1)
**bio2** — **Mean diurnal range (monthly temp. range)**
**AHM** — **Annual heat moisture index**
SHM — Summer heat moisture index
CMD — Hargreaves climatic moisture index
bio8 — Mean temp. of wettest quarter
**bio5** — **Max temp. of warmest month**
EXT — Extreme max. temp. over 30 years
MAP — Mean annual precipitation
bio12 — Annual precipitation
bio19 — Precipitation of coldest quarter
PPT_wt — Winter precipitation
bio14 — Precipitation of driest month
**bio17** — **Precipitation of driest quarter**
bio15 — Precipitation seasonality
DD_0 — Degree–days below 0°C
bio7 — Temp. annual range
**bio4** — **Temp. seasonality**
TD — Difference between MCMT and MWMT
bio9 — Mean temp. of driest quarter
bio6 — Min. temp. of coldest month
EMT — Extreme min. temp. over 30 years
bio18 — Precipitation of warmest quarter
MSP — Mean summer precipitation
PPT_sm — Summer precipitation
bio13 — Precipitation of wettest month
**bio16** — **Precipitation of wettest quarter**

Common Garden Site IDs

**d** *Neural network-based phenotypic classification*

Ecotype
- Coastal
- Lowland
- Upland

Upland leaf/tiller traits
LDA Axis #1 (78.16%)
LDA Axis #2 (18.04%)
Upland whole-plant/color morphotype

**e** *Multivariate expert ecotype assignment in the TX2 garden*

Ordinal ecotype assignment 1 = Lowland, 5 = Upland

Leaf | Tiller | Whole Plant/Color | Overall

**f** *LDA-loadings of 32 phenotypes*

**g** *Description of phenotypes*

| Variable ID | MI | TX2 |
|---|---|---|
| leaf length | 1.3 | 1.2 |
| leaf width | 1.7 | 3.5 |
| width:length (leaf) | 1.5 | 3.9 |
| tiller:leaf (height) | 1.4 | 4.8 |
| panicle emergence (date) | 4.7 | 1.6 |
| plant size (tillers x height) | 3.3 | 3.1 |
| canopy height | 3.2 | 4.3 |
| green-up (date) | 4.1 | 3.7 |
| n. tillers (end-of-season) | 4.2 | 3.6 |
| lamina:midrib (thickness) | 2.1 | 6.1 |
| leaf area | 3.8 | 4.6 |
| lamina thickness | 3.1 | 5.3 |
| height:count (panicle) | 4.5 | 4.12 |
| n. tillers (first cohort) | 4.9 | 4.11 |
| height:count (leaf) | 4.1 | 5.1 |
| panicle height | 5.2 | 4.4 |

**h** *Association between biomass and climate-distance*

$r^2$ from quadratic model

Climatic variable: MAT | BIO5 | BIO4 | BIO17 | AHM | BIO2 | BIO16

**Extended Data Fig. 2** | See next page for caption.

**Extended Data Fig. 2 | Phenotypic and climatic gradients among common gardens and ecotypes. a**, The ten common gardens span much of the geographical distribution of, and elicit very different phenotypic responses among, our switchgrass diversity panel. For each garden, we present the georeferenced location and some basic quantitative genetic attributes of the plants grown there. **b**, To illustrate the climate context of winter mortality, we present a seven-day rolling mean of minimum daily temperature across the study period. Line colours match the colour key in **a**. **c**, To investigate the climatic attributes of each garden, we clustered 46 climatic variables from WorldClim (variables are named bio1–19[22]) and ClimateNA[21] using the georeferenced locations for the diversity panel; the identifiers (left) and description (right) accompany each row. These seven clusters, separated by breaks in the heat map, are represented by the seven climate variables that most closely correlated with the first principal component eigenvector of each cluster (labelled in bold). **d**, To investigate ecotype evolution, we probabilistically assigned each member of the diversity panel to one of three ecotypes ($n_{upland} = 221$, $n_{coastal} = 157$, $n_{lowland} = 129$) using a set of morphological

($n = 16$ at 2 gardens) and qualitative ($n = 2$) phenotypes; the linear discriminant functions that distinguish the ecotypes are presented here along with the eigenvectors of the two qualitative ecotype categorizations. Each point represents a single genotype grown in both $TX_2$ and MI gardens ($n = 509$). LDA, linear discriminant analysis. **e**, Qualitative ecotype assessments from experts are presented for the $TX_2$ garden in 2019. The $y$-axis scale is ordinal with five categories, but points are jittered so that the density of observations is more obvious. Points are coloured by neural network classification following **d**. **f**, Loadings for the other 16 variables (across 2 gardens) are plotted on the same scale and axes as **d**. To distinguish variables, we clustered each into one of four groups, representing variation in leaf (dark green) (3), whole plant (red) (1) and combinations of these. **g**, The table presents a legend for the labels in **f**, in which each variable was measured in both MI and $TX_2$ gardens. More detailed descriptions of the phenotypes can be found in Supplementary Data 5. **h**, For each of the seven climate variables, we corrected climate distance between the collection site and each common garden. The quadratic model fit ($r^2$) for each variable and ecotype are presented.

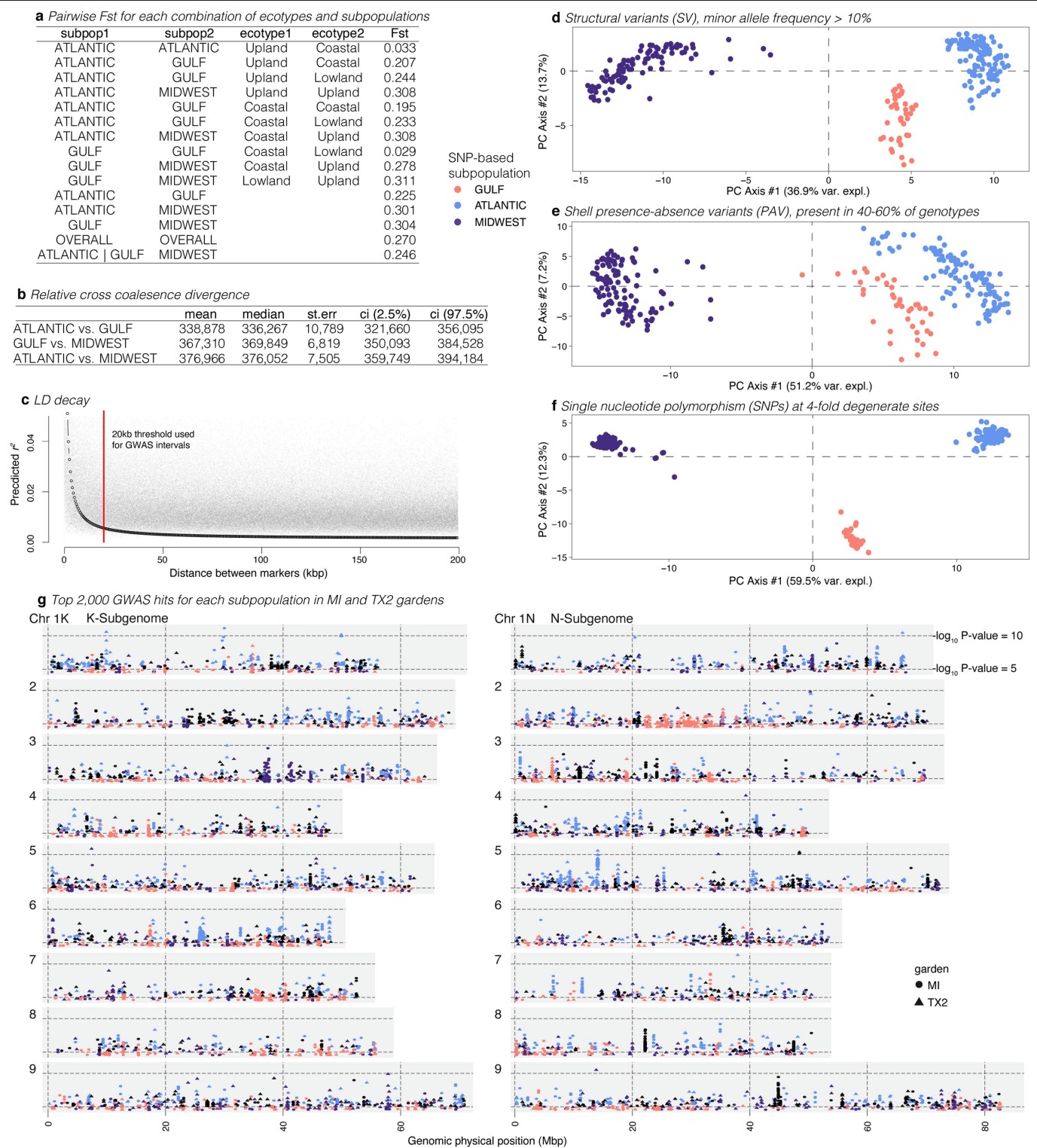

**a** *Pairwise Fst for each combination of ecotypes and subpopulations*

| subpop1 | subpop2 | ecotype1 | ecotype2 | Fst |
|---|---|---|---|---|
| ATLANTIC | ATLANTIC | Upland | Coastal | 0.033 |
| ATLANTIC | GULF | Upland | Coastal | 0.207 |
| ATLANTIC | GULF | Upland | Lowland | 0.244 |
| ATLANTIC | MIDWEST | Upland | Upland | 0.308 |
| ATLANTIC | GULF | Coastal | Coastal | 0.195 |
| ATLANTIC | GULF | Coastal | Lowland | 0.233 |
| ATLANTIC | MIDWEST | Coastal | Upland | 0.308 |
| GULF | GULF | Coastal | Lowland | 0.029 |
| GULF | MIDWEST | Coastal | Upland | 0.278 |
| GULF | MIDWEST | Lowland | Upland | 0.311 |
| ATLANTIC | GULF | | | 0.225 |
| ATLANTIC | MIDWEST | | | 0.301 |
| GULF | MIDWEST | | | 0.304 |
| OVERALL | OVERALL | | | 0.270 |
| ATLANTIC \| GULF | MIDWEST | | | 0.246 |

SNP-based subpopulation
- GULF
- ATLANTIC
- MIDWEST

**b** *Relative cross coalesence divergence*

| | mean | median | st.err | ci (2.5%) | ci (97.5%) |
|---|---|---|---|---|---|
| ATLANTIC vs. GULF | 338,878 | 336,267 | 10,789 | 321,660 | 356,095 |
| GULF vs. MIDWEST | 367,310 | 369,849 | 6,819 | 350,093 | 384,528 |
| ATLANTIC vs. MIDWEST | 376,966 | 376,052 | 7,505 | 359,749 | 394,184 |

**c** *LD decay*

20kb threshold used for GWAS intervals

**d** *Structural variants (SV), minor allele frequency > 10%*

**e** *Shell presence-absence variants (PAV), present in 40-60% of genotypes*

**f** *Single nucleotide polymorphism (SNPs) at 4-fold degenerate sites*

**g** *Top 2,000 GWAS hits for each subpopulation in MI and TX2 gardens*

Chr 1K  K-Subgenome

Chr 1N  N-Subgenome

$-\log_{10}$ P-value = 10
$-\log_{10}$ P-value = 5

garden
- MI
- TX2

Genomic physical position (Mbp)

**Extended Data Fig. 3 | Population and quantitative genetic divergence between and evolution within subpopulations and ecotypes. a**, Pairwise *F*-statistics between each subpopulation-by-ecotype combination and across all ecotypes for each subpopulation. **b**, Cross coalescence (RCCR) represents an alternative method to define divergence. Here, 16 bootstraps of RCCR profiles were converted to generation time at which divergence occurred. Statistics across the bootstraps are presented. **c**, Linkage-disequilibrium nonlinear function of physical distance and predicted correlation coefficients among markers for the entire sample. The linear model prediction for each 500-bp interval is plotted as black open points; 2-bp-interval mean $r^2$ values are the light grey points in the background. **d**–**f**, Population genetic structure is displayed as the principal coordinates from a scaled and centred distance matrix of structural variants (**d**), presence–absence variants (**e**) and SNPs (**f**), colour-coded by subpopulation assignments in Fig. 3. **g**, Positions and $-\log_{10}(P$ values) of the top 2,000 GWAS hits are presented for 2 gardens, the 3 subpopulations (coloured as in **d**–**f**) and an overall run (black points).

**a** *Subgenome bias of biomass heritability (N-K) by subpopulation and garden*

| garden | GULF | ATLANTIC | MIDWEST | OVERALL |
|---|---|---|---|---|
| SD | | 0.107 | 0.048 | |
| MI | 0.01 | 0.139 | 0.06 | |
| IL | 0.044 | | | 0.58 |
| NE | 0.026 | 0.091 | 0.118 | |
| MO | 0.004 | 0.021 | 0.057 | |
| OK | 0.066 | 0.013 | 0.31 | |
| TX4 | -0.088 | 0.019 | | 0.049 |
| TX3 | 0.071 | 0.016 | -0.027 | 0.037 |
| TX2 | -0.032 | 0.087 | 0.055 | 0.012 |
| TX1 | 0.002 | | | 0.054 |

**c** *Subgenome counts across 26 tests*

| ID | K | N | K bias |
|---|---|---|---|
| n. variable SNPs (ATLANTIC) * | 4,938,797 | 5,661,410 | 0.068 |
| n. variable SNPs (GULF) * | 4,816,319 | 5,503,882 | 0.067 |
| n. variable SNPs (MIDWEST) * | 4,093,574 | 4,730,765 | 0.072 |
| n. variable SNPs (Overall)* | 4,727,783 | 5,456,073 | 0.072 |
| n. sig. GWAS hits (climate ATLANTIC) | 2,373 | 2,139 | 0.052 |
| n. sig. GWAS hits (climate GULF) | 989 | 1,128 | -0.066 |
| n. sig. GWAS hits (climate MIDWEST) | 2,716 | 2,559 | 0.030 |
| n. sig. GWAS hits (climate Overall) | 6,076 | 5,824 | 0.021 |
| n. sig. GWAS hits (fitness ATLANTIC) | 115 | 117 | -0.009 |
| n. sig. GWAS hits (fitness GULF) | 5 | 4 | 0.111 |
| n. sig. GWAS hits (fitness MIDWEST) | 54 | 40 | 0.149 |
| n. sig. GWAS hits (fitness Overall) | 174 | 161 | 0.039 |
| n. upregulated genes (Overall) | 6,123 | 5,133 | 0.088 |
| n. upregulated genes (MI) | 5,315 | 4,402 | 0.094 |
| n. upregulated genes (TX2) | 5,445 | 4,477 | 0.098 |
| n. total annotated genes | 40,957 | 38,712 | 0.028 |
| n. present genes (1:1 PAV with outgroups) | 1,419 | 497 | 0.481 |
| mean synonymous subst. rate (ks)* | 0.092 | 0.096 | 0.021 |
| mean 4-fold tr ratio (4dtv)* | 0.049 | 0.053 | 0.034 |
| mean non-synon. subst. rate (ka)* | 0.029 | 0.03 | 0.022 |
| total heritability (Atlantic) | 0.236 | 0.73 | -0.511 |
| total heritability (Overall) | 0.079 | 0.811 | -0.823 |
| total heritability (GULF) | 0.409 | 0.511 | -0.111 |
| total heritability (MIDWEST) | 0.186 | 0.807 | -0.625 |
| bp. introgressions (MIDWEST to ATLANTIC) | 53,647,121 | 61,626,765 | -0.069 |
| n. introgressions (MIDWEST to ATLANTIC) | 1,879 | 2,235 | -0.087 |

*Variables where elevated values of N indicated K bias. These include the number of variable sites and the rate of nucleotide substitutions, since stronger purifying selection should act to reduce these.*

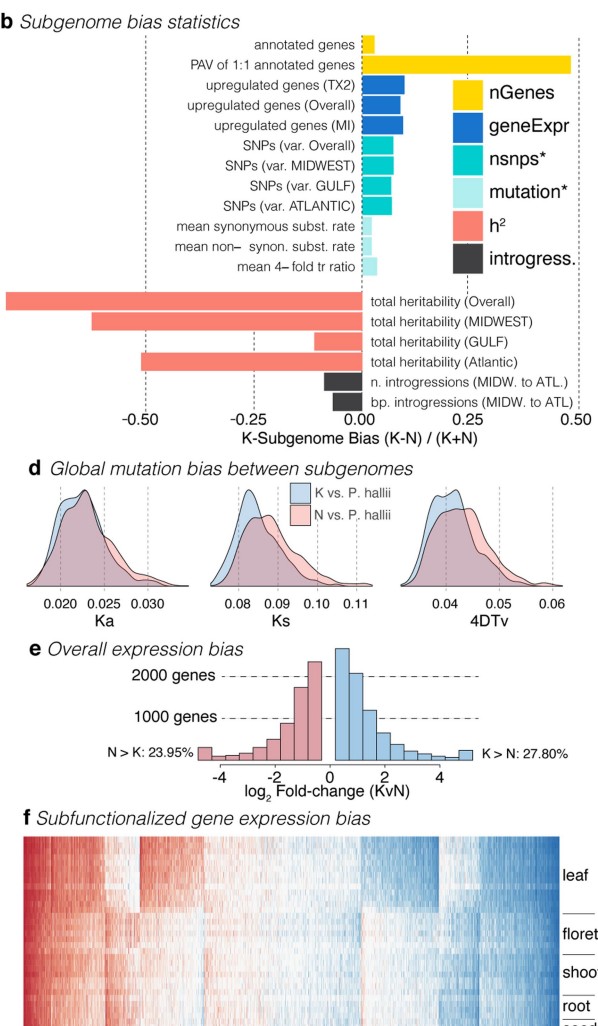

**b** *Subgenome bias statistics*

**d** *Global mutation bias between subgenomes*

**e** *Overall expression bias*

**f** *Subfunctionalized gene expression bias*

**Extended Data Fig. 4 | Subgenome biases across DNA, expression and quantitative traits. a**, Difference in biomass SNP–heritability ($h^2$) estimates between subgenomes for each garden-by-subpopulation combination. Garden-by-subpopulation combinations with empty cells indicate that the model did not converge. **b**, Subgenome bias for all sets of genome analyses conducted here. Colours indicate the dataset used. **c**, Counts and ratios used to build **b**, with longer descriptions of the variables. **d**, Density distributions of nonsynonymous ($K_a$), synonymous ($K_s$) and fourfold-degenerate transversion substation rates (4DTv) for each subgenome relative to *P. hallii*. **e**, Summation of the number of genes in each colour bin of **f**. **f**, A heat map of expression in which K > N (blue) and N > K (red) is shown for each tissue in the genome-annotation RNA-seq dataset.

## Extended Data Table 1 | Heritability due to SNPs and background kinship

| subpopulation | response | SNP-$h^2$ (± SE) | polygenic background-$h^2$ (± SE) |
|---|---|---|---|
| Atlantic | biomass_CLMB | 56.67±22.6 | 22.64±7.3 |
| Atlantic | biomass_KBSM | 57.87±7.8 | 7.79±6.3 |
| Atlantic | biomass_PKLE | 55.53±7.5 | 7.49±5.3 |
| Atlantic | AHM | 38.7±8.8 | 8.79±4.3 |
| Atlantic | bio2 | NA | NA |
| Atlantic | bio4 | 82.91±17.1 | 17.09±7.6 |
| Atlantic | bio5 | 64.31±33.7 | 33.73±10.4 |
| Atlantic | bio16 | 80.81±19.1 | 19.09±6.9 |
| Atlantic | bio17 | 94.13±4.9 | 4.88±2.9 |
| Atlantic | MAT | 39.74±13.5 | 13.53±4.6 |
| Gulf | biomass_CLMB | 56.81±1.2 | 1.16±16.5 |
| Gulf | biomass_KBSM | 31.66±11.2 | 11.25±12.7 |
| Gulf | biomass_PKLE | 27.84±23.3 | 23.35±13.2 |
| Gulf | AHM | 57.42±39.8 | 39.79±17.8 |
| Gulf | bio2 | 51.15±47.7 | 47.65±19.9 |
| Gulf | bio4 | 69.03±31 | 30.97±19.6 |
| Gulf | bio5 | 79.67±18.3 | 18.26±18 |
| Gulf | bio16 | 38.66±61.3 | 61.29±15 |
| Gulf | bio17 | 57.35±42.6 | 42.57±17 |
| Gulf | MAT | 58.15±1.1 | 1.06±11.2 |
| Midwest | biomass_CLMB | 48.12±28.9 | 28.91±10.9 |
| Midwest | biomass_KBSM | 67.72±32.3 | 32.28±6.4 |
| Midwest | biomass_PKLE | 61.53±29 | 28.98±6.7 |
| Midwest | AHM | 66.67±9.4 | 9.42±2.5 |
| Midwest | bio2 | 96.34±2.8 | 2.76±1.6 |
| Midwest | bio4 | 69.03±31 | 30.97±19.6 |
| Midwest | bio5 | 98.17±1.4 | 1.38±0.8 |
| Midwest | bio16 | 98.29±1.6 | 1.6±1 |
| Midwest | bio17 | 97.78±2.2 | 2.15±1.2 |
| Midwest | MAT | 72.03±2.1 | 2.06±1.1 |
| Full | biomass_CLMB | 49.44±6.3 | 6.3±5.8 |
| Full | biomass_KBSM | 44.04±0.2 | 0.21±4.7 |
| Full | biomass_PKLE | 35.93±9.1 | 9.08±4.5 |
| Full | AHM | 94.92±0.6 | 0.6±10.8 |
| Full | bio2 | 77.9±20.8 | 20.82±12.1 |
| Full | bio4 | 50.03±50 | 49.97±12.3 |
| Full | bio5 | 36.65±62.2 | 62.24±10.5 |
| Full | bio16 | 76.95±23 | 22.99±11.8 |
| Full | bio17 | 90.96±9 | 8.95±7.5 |
| Full | MAT | 41.46±33.3 | 33.32±9.1 |

Heritability of traits and climate-of-origin variation was partitioned to SNPs in GWAS hits ('SNP–$h^2$') and background or polygenic variation. SNP–heritability and standard errors are presented for each of seven climate variables, biomass in three gardens within and across ('full') each of the subpopulations. Response variable-by-subpopulation combinations marked with 'NA' indicate that the model did not converge.

# nature research

# Reporting Summary

Nature Research wishes to improve the reproducibility of the work that we publish. This form provides structure for consistency and transparency in reporting. For further information on Nature Research policies, see our Editorial Policies and the Editorial Policy Checklist.

## Statistics

For all statistical analyses, confirm that the following items are present in the figure legend, table legend, main text, or Methods section.

| n/a | Confirmed | |
|---|---|---|
| ☐ | ☒ | The exact sample size (*n*) for each experimental group/condition, given as a discrete number and unit of measurement |
| ☒ | ☐ | A statement on whether measurements were taken from distinct samples or whether the same sample was measured repeatedly |
| ☐ | ☒ | The statistical test(s) used AND whether they are one- or two-sided *Only common tests should be described solely by name; describe more complex techniques in the Methods section.* |
| ☐ | ☒ | A description of all covariates tested |
| ☐ | ☒ | A description of any assumptions or corrections, such as tests of normality and adjustment for multiple comparisons |
| ☐ | ☒ | A full description of the statistical parameters including central tendency (e.g. means) or other basic estimates (e.g. regression coefficient) AND variation (e.g. standard deviation) or associated estimates of uncertainty (e.g. confidence intervals) |
| ☐ | ☒ | For null hypothesis testing, the test statistic (e.g. *F*, *t*, *r*) with confidence intervals, effect sizes, degrees of freedom and *P* value noted *Give P values as exact values whenever suitable.* |
| ☐ | ☒ | For Bayesian analysis, information on the choice of priors and Markov chain Monte Carlo settings |
| ☐ | ☒ | For hierarchical and complex designs, identification of the appropriate level for tests and full reporting of outcomes |
| ☐ | ☒ | Estimates of effect sizes (e.g. Cohen's *d*, Pearson's *r*), indicating how they were calculated |

*Our web collection on statistics for biologists contains articles on many of the points above.*

## Software and code

Policy information about availability of computer code

| Data collection | DNA and RNA data were collected via Illumina and PacBio internal routines. All phenotype data was collected and input manually using best practices for quantitative genetics data. |
|---|---|
| Data analysis | All data analysis was conducted through programs described in the methods. The majority of which was accomplished in the R environment for statistical computing. Other programs included: ancestry_hmm, vcftools, bcftools, samtools, varscan, PLINK, SHAPEIT, bwa-mem, Picard, GATK, MSMC, GSNAP, HTSeq, Dialign-TX, Gblocks, mafft, orthofinder, Repeatmasker, RepeatModeler, PASA, EXONERATE, Jellyfish, LTRHarvest, MECAT, BLAT, ARROW, and FlowJo. The GWAS pipeline developed here can be found on github (see code availability statement). |

For manuscripts utilizing custom algorithms or software that are central to the research but not yet described in published literature, software must be made available to editors and reviewers. We strongly encourage code deposition in a community repository (e.g. GitHub). See the Nature Research guidelines for submitting code & software for further information.

## Data

Policy information about availability of data

All manuscripts must include a data availability statement. This statement should provide the following information, where applicable:
- Accession codes, unique identifiers, or web links for publicly available datasets
- A list of figures that have associated raw data
- A description of any restrictions on data availability

SRA accession codes for all RNA and DNA sequencing libraries can be found in Supplemental Data 3 and 4 respectively. The v5 AP13 genome has be deposited at DDBJ/ENA/GenBank under the accession JABWAI000000000. The genome, gene and repeat annotations can also be downloaded directly from Phytozome: https://phytozome-next.jgi.doe.gov/info/Pvirgatum_v5_1. With the exception of map layers (Fig. 2a, Fig. 3a), which are publicly available from naturalearth.org, raw data

for all figures can be found in the source data file or associated tables in the extended data and supplementary material.

# Field-specific reporting

Please select the one below that is the best fit for your research. If you are not sure, read the appropriate sections before making your selection.

☐ Life sciences　　　☐ Behavioural & social sciences　　　☒ Ecological, evolutionary & environmental sciences

For a reference copy of the document with all sections, see nature.com/documents/nr-reporting-summary-flat.pdf

# Ecological, evolutionary & environmental sciences study design

All studies must disclose on these points even when the disclosure is negative.

| | |
|---|---|
| Study description | We conducted quantitative analysis of phenotypes, collected in common gardens, and gene expression, collected in both common gardens and controlled conditions in the lab. |
| Research sample | Panicum virgatum (switchgrass) plants represent the entirety of the study design. Individual genotypes were clonally replicated and phenotyped in the field. Leaf and other tissue was assayed for gene expression. Replicated measures on a single individual plant were collapsed to a breeding value and never used as a unit of replication in any analyses. |
| Sampling strategy | Phenotyping was always conducted in a completely randomized design within blocks (common gardens). |
| Data collection | Field data were collected by a team of field technicians. The identities of the field techs always accompanied the measurements and care was taken to ensure that no systematic biases resulted from field technician factors. |
| Timing and spatial scale | Sample size was determined as a function of field experimental restrictions and sequencing cost. We sequenced 732 genotypes to maximize diversity within a limited budget. These plants were grown in as many sites as possible. For some sites, there was not enough space to grow all plants. In these cases, we chose plants that (a) represented the maximum genetic diversity and (b) had enough clonal replicates available. |
| Data exclusions | We discuss the libraries excluded in the methods. Some libraries were excluded due to poor sequencing quality or likely contamination. |
| Reproducibility | Plants were grown as clonal replicates. We opted for this approach (in lieu of full/half sib designs) to maximize repeatability: the exact same genotypes can be grown in other experiments. |
| Randomization | At each garden, planting was completely randomized in a single block. |
| Blinding | All field experiments were conducted using genotype identifiers that do not have an obvious connection to the location, name, etc. of each genotype. The anonymous 4- or 5-digit 'Library ID' was used for all statistical genomic analyses. It is impossible to conduct analyses blind of these identifiers, since all data is entered and output along with the IDs; however, we took care to use only these anonymous IDs and without direct reference to their biological names or context. |

Did the study involve field work?　☒ Yes　　☐ No

## Field work, collection and transport

| | |
|---|---|
| Field conditions | Field conditions were ambient at 10 common gardens over two years. Daily rainfall, temperature and soil conditions can be made available, but represent far too much data to place in this document. Summary climate data can be found in extended data figure 2. |
| Location | Here are the georeferenced coordinates of the 10 common gardens:<br>BRKG: 44.30680(lat), -96.67050(lon)<br>CLMB: 38.89690(lat), -92.21780(lon)<br>FRMI: 41.83671(lat), -88.23960(lon)<br>KBSM: 42.41962(lat), -85.37127(lon)<br>KING: 27.54986(lat), -97.88101(lon)<br>LINC: 41.15430(lat), -96.41530(lon)<br>OVTN: 32.30290(lat), -94.97940(lon)<br>PKLE: 30.38398(lat), -97.72938(lon)<br>STIL: 35.99115(lat), -97.04649(lon)<br>TMPL: 31.04338(lat), -97.34950(lon) |
| Access & import/export | All plant collections were conducted either from established agricultural gardens under the managers permission, or from collaborators under their own collecting permits. |
| Disturbance | Collections of natural habitats were conducted with the utmost care by professional botanists following protocols outlined in the collection permits. Common garden field sites were always constructed in previously disturbed or agricultural lands. |

# Reporting for specific materials, systems and methods

We require information from authors about some types of materials, experimental systems and methods used in many studies. Here, indicate whether each material, system or method listed is relevant to your study. If you are not sure if a list item applies to your research, read the appropriate section before selecting a response.

## Materials & experimental systems

| n/a | Involved in the study |
|---|---|
| ☒ | Antibodies |
| ☒ | Eukaryotic cell lines |
| ☒ | Palaeontology and archaeology |
| ☒ | Animals and other organisms |
| ☒ | Human research participants |
| ☒ | Clinical data |
| ☒ | Dual use research of concern |

## Methods

| n/a | Involved in the study |
|---|---|
| ☒ | ChIP-seq |
| ☐ | ☒ Flow cytometry |
| ☒ | MRI-based neuroimaging |

## Flow Cytometry

### Plots

Confirm that:

☒ The axis labels state the marker and fluorochrome used (e.g. CD4-FITC).

☒ The axis scales are clearly visible. Include numbers along axes only for bottom left plot of group (a 'group' is an analysis of identical markers).

☒ All plots are contour plots with outliers or pseudocolor plots.

☒ A numerical value for number of cells or percentage (with statistics) is provided.

### Methodology

| | |
|---|---|
| Sample preparation | 200-300 mg of young leaf tissue was macerated in a petri dish with a razor blade and treated for 15 minutes with 1mL Cystain PI Absolute P nuclei extraction buffer (Sysmex Flow Cytometry) mixed with 1μL 2-mercaptoethanol. Samples were then filtered to isolate free nuclei with a CellTrics 30 μm filter (Sysmex) and treated for 20 minutes on wet ice with 2mL of Cystain PI Absolute P staining buffer (Sysmex), 12μL of propidium iodide and 6μL of RNase A |
| Instrument | LSRFortessa SORP Flow Cytometer (BD Biosciences) |
| Software | FlowJo software (BD Biosciences) |
| Cell population abundance | NA |
| Gating strategy | Samples were binned into three categories based upon the average units of fluorescence per nuclei. Ploidy level of the sample was considered 4X if the cell population had 40-80K units of fluorescence, 6X for 80-100K units and 8X for 100-140K units. The binning parameters were established with flow cytometry data from several P. virgatum accessions of known ploidy. T |

☒ Tick this box to confirm that a figure exemplifying the gating strategy is provided in the Supplementary Information.

