## [Peer Review File · Nature]

Peer Review File**Manuscript Title:** Genomic mechanisms of climate adaptation in polyploid bioenergy switchgrass**Editorial Notes:****Reviewer Comments & Author Rebuttals****Reviewer Reports on the Initial Version:**

Referees' comments:

Referee #1 (Remarks to the Author):

In the manuscript entitled "Polyploidy and genomic introgressions facilitate climate adaptation and biomass yield in switchgrass" Lovell et al., present an impressive dataset comprised of a high-quality switchgrass genome, and a large set of ecotypes (732) that they re-sequenced and phenotyped across 10 common garden sites across the USA. The tetraploid genome is chromosome scale and resolved into sub-genomes, which the authors leverage to generate a SNP diversity map across three-types of tetraploid ecotypes. The authors leverage machine learning to categorize the ecotypes into phenotypic classes to deconvolute genetic background, site of collection and growth response across the common garden experiment. The authors develop a GWAS methodology to handle the sheer volume of data generated and demonstrate that switchgrass has extensive and differential loci controlling biomass and climate of origin fitness. Next the authors looked specifically at putative introgressions and found that Midwest contributed to the Atlantic subpopulations and that this accounted for climatic adaption of the former population. Finally, the authors present an interesting paradox where the K sub-genome shows dominance, yet the N sub-genome shows more biomass SNP heritability and introgressions.

The authors expertly present a phenomenal amount of complex data with both attractive and informative figures. The writing is clear and concise, although at times could be a bit more descriptive in both the main text and methods. For instance, many tools are cited and only sometimes explained what they do; while some reviewers/reader may be familiar with some of these tools, they most likely are not familiar with all. Also impressive is the rigor of the analyses, which are well designed with accompanying comprehensive statistical support. One criticism is that the title is basically a truism-most people already agree that polyploidy and introgression facilitate climate adaptation-hence the term ecotype. There are many recent reviews in covering both topics that despite the extensive reference list the authors don't include to provide context to their finding. What the authors do bring to light is the extent of the introgressions and polyploid paradox in a mostly undomesticated yet potentially economically important bioenergy "crop." For instance, what makes this study more than an enormous amount of data and a high-quality genome that support what most breeders/geneticists/ecologists already believe (although I agree that believing and proving are quite different)? To that end, one of the potentially missed opportunities or confounding features is that the authors only look at SNPs. While the panicum genomes (and most grass genomes) are highly colinear, questions of polyploidy (fractionation), sub-genome dominance, and introgression all beg the question of the variation that is not in the reference genome. While the methods are state of the art or better in terms of the variation analysis, the authors do have an incredible amount of Illumina sequencing (median=59x) that could be used for de novo assembly and gene content analysis (despite the complexity of the switchgrass genome). Also, this reviewer was left hanging with the excitement as to what genes/loci/variation were underlying the GWAS and introgression peaks. Understandably it would be highly speculative, and some may say descriptive, although since the loci are differential across

the GWAS peaks there may be commonalities that are counterintuitive much like the K/N sub-genome dominance paradox.

Below are some specific suggestions.

Line 95: "...large, repetitive, polyploid and heterozygous..." What distinguishes switchgrass in this sentence? Wheat is much larger and more repetitive, and higher ploidy. Would be good to put this comment in context because 1.1 Gb (or 2.2 Gb) is not that large of a genome. Also, the level of heterozygosity is not mentioned.

Line 97: define "complete" and how did you establish this?

Line 98: N50=5.5M base pairs, awkward wording-maybe just spell it all out: "megabases." Wasn't the N50 initially 1.1 Mb in the methods (line 439)? Also, is it N50 or L50? Different between the two places.

Line 107-108: The use of the wording "for the first time," really requires something special. A haplotype and sub-genome resolved tetraploid would merit a statement like this. How does this assembly compare to the two closely related tetraploid broomcorn millet genomes (*Panicum miliaceum*)?

Line 125-126: why not de novo assemble and anchor to the contigs to look at the later mentioned introgressed regions? Why not do a pan genome analysis with this great number of ecotypes? Are there regions that are unique to the resequencing that cannot be accessed other than by assembly?

Unless I missed it (which is possible since there was a lot of information and inaccessible content) the phenotypic data was not described. The predictive methods were explained but not how the measurements were taken etc. Or is the second paragraph in the first section the "phenotyping" section? If so, where is the description of how leaf phenotypes were taken (lines 131-135).

Lines 201-206: Are the usual suspects found under the loci for climate and fitness. Having a high-quality reference and gene prediction (and RNAseq) provide an opportunity to confirm known or even better yet identify loci previously not associated with these traits. Since there was minimal overlap across the identified regions it would be interesting if the loci are from similar gene ontology categories or expression network modules; ie different genes but similar pathways.

Lines 218-222: What are the locations of the overlap between the GWAS and introgressions? Seems like this should be a very interesting gene/pathway list.

Lines 233-236: What was the relative gene retention in the introgression regions? This is where de novo assembling the ecotypes and anchoring them to the introgressed regions may reveal interesting present/absence variation.

Lines 263-265: Is it possible that sub-genome K might have variation not detected by SNP/resequencing analysis?

Lines 269-284: The discussion could place the finding in the context of more recent reviews/results across wild plants and crops (as mentioned above). It would be interesting get more insight into the significance of having a less dominant genome having the introgressions and important variation.

Figure 4c, why are the dots in the ocean?

Extended Data Figure 1a what is the basis of the claim that this genome is among the best

available plant genomes? Would be great, especially in the extended data, to give the reader some type of gauge for how difficult a plant genome and the accomplishments with the switchgrass genome compared to what is currently out there. The authors have done such a great job with providing the community with high quality genomes in the past, this is a great opportunity to highlight it with more than a throw away sentence in a legend.

The extended data tables were not openable (looked like binary or some other format). Downloaded several times to check. Could not review the content.

Referee #2 (Remarks to the Author):

The attached file contains these comments with formatting that makes them easier to read.

I used the guide for reviewers to organize my comments to the authors and editors below.

Key results: Please summarise what you consider to be the outstanding features of the work.

I found this manuscript to be outstanding for the following reasons.

- It uses novel methods to present novel insights on topics of broad interest and societal importance. Specifically, the authors coupled multiple lines of evidence (genomic markers, plant morphology and performance, weather indicators) to show not only that switchgrass is doing surprising things with implications for our understanding of plant evolution, but that plant breeders might be going about improving plants in the wrong way to rapidly develop crops in a changing climate.

- The methods with which I am most familiar in this paper, the common garden and weather modeling work, are top-notch and frankly boggle the mind. Pulling off a study of this magnitude is herculean but is exactly what was needed to reveal holistic patterns in plant form, function, genetics, survival, and heritability that have been limiting perennial grass improvement. This study avoids the limitations of many other studies by measuring the things that matter in the locations that matter; it isn't a snapshot of performance in a given location, nor is it a meta-analysis with confounding variables.

- One of the coolest things about this study is the insight that non-dominant sub-genomes may be a repository for survival genes, and that a benefit of polyploidy is the relaxed regulation of the sub-genome that allows genes that may not confer fitness at the moment to persist until conditions arise in which they do confer fitness. I took away from this study that the sub-genome can act like your attic or your basement: you rarely clean it, but there are things there that might be needed someday.

- I found the discussion in this paper to be supported by the evidence presented and written to clearly advance science. Particularly salient was the authors' observation that science's current tendency to avoid complex systems like obligate outcrossing, polyploidy species in favor of simpler model systems might be misguided for plant improvement in a changing climate.

Validity: Does the manuscript have flaws which should prohibit its publication? If so, please provide details.

Before publication I would like to see details of the morphometric scales the team used to measure plant morphology and ecotypes across genotypes and common gardens. To my knowledge, there is no agreed-upon, systemic scale available to the scientific community to measure switchgrass features consistently. This work is not repeatable if we don't know how they measured their plants. Further, it was not clear how morphometric data was used to determine ecotype differences. I would like to see a supplement that gives repeatable instructions for how morphology was measured, and how metrics led to categorization as upland, lowland, or coastal. I believe such a supplement will greatly increase the citation rate for this paper by providing a scale and methods the rest of the perennial grass community can use.

Originality and significance: If the conclusions are not original, please provide relevant references.

On a more subjective note, do you feel that the results presented are of immediate interest to many people in your own discipline, and/or to people from several disciplines?

I believe the results are original. I am fully confident that the results will be strong interest to the

plant evolution and ecology disciplines, and of broad interest to those interested in rapid adaptation of species to different growth conditions.

Data & methodology: Please comment on the validity of the approach, quality of the data and quality of presentation. Please note that we expect our reviewers to review all data, including any extended data and supplementary information. Is the reporting of data and methodology sufficiently detailed and transparent to enable reproducing the results?

Please see previous comments about ecotypes and reproducible guides to morphology methods and metrics.

Appropriate use of statistics and treatment of uncertainties: All error bars should be defined in the corresponding figure legends; please comment if that's not the case. Please include in your report a specific comment on the appropriateness of any statistical tests, and the accuracy of the description of any error bars and probability values.

I believe the statistical tests to be appropriate.

Conclusions: Do you find that the conclusions and data interpretation are robust, valid and reliable?

Yes.

Suggested improvements: Please list additional experiments or data that could help strengthening the work in a revision.

Data indicating how ecotypes were determined based on morphometric data would strengthen this paper.

References: Does this manuscript reference previous literature appropriately? If not, what references should be included or excluded?

Yes.

Clarity and context: Is the abstract clear, accessible? Are abstract, introduction and conclusions appropriate?

Yes.

Please indicate any particular part of the manuscript, data, or analyses that you feel is outside the scope of your expertise, or that you were unable to assess fully.

I am not fully qualified to assess the genomic assembly and annotation.

Please address any other specific question asked by the editor via email.

None.

Referee #3 (Remarks to the Author):

Summary

In this work, the authors present a version 5 assembly and annotation of the switchgrass genome, 'AP13', which is characterized by several complexities including its large size, high degree of heterozygosity, and polyploid and repetitive nature. One marked improvement in this v5 assembly and annotation was the collinearity between the N and K sub-genomes. Methodology presented here would be relevant for other complex genomes, most especially for other polyploid plant species. The authors go on to investigate the genetic basis of climate adaptation, by assembling and re-sequencing a diversity panel of 732 tetraploid switchgrass genotypes, which were phenotyped for survival and biomass production, among other traits (morphology, phenology, etc) in 10 common gardens that span 1862km of latitude. Ecotypes were assigned to the majority of the panel, and these assignments were also included as variables for subsequent analyses. The main conclusion from the survival and biomass analyses was that accessions tend to perform best in climates that are most similar to where they originated, and that climate-of-origin variables related to temperature, especially 30-year minimum temperature, were the best predictors of survival in the common garden trials. Population structure of the diversity panel was next investigated and compared to ecotype assignments; the primary finding here is that ecotypes and subpopulations were not as aligned as previously assumed. The authors also perform climate and biomass GWAS to understand the contribution of loci from each sub-genome to climate adaptation and further investigated whether the prevalence of upland ecotypes in both MIDWEST and

ATLANTIC genetic subpopulations was due to independent origins or admixture with a substantial set of analyses. They discovered that MIDWEST introgressions into northern ATLANTIC accessions were more likely to contain significant GWAS intervals shared between the two subpopulations, supporting the hypothesis that introgression from MIDWEST facilitated northern expansion of the ATLANTIC subpopulation.

Overall, the manuscript is well-written and the presentation logical. It is clear that a substantial amount of work went into the study. The new assembly and annotation of AP13 along with the re-sequenced diversity panel is sure to provide the switchgrass community with a valuable new genetic resource whose benefits will continue to be realized for a long time to come. With respect to the evolutionary and ecology questions addressed, the finding that accessions tend to perform best under conditions most similar to the climate from where they originate is not novel. However, the careful dissection of the potential contribution of introgressions to facilitate subpopulation range expansion was well done. The weakest part of the study, in my opinion, is related to ecotype assignment and the general treatment of trait data. These are outlined below.

Specific Comments

1. Concerns related to ecotype assignment. In the Methods, it states that only seven cultivars were used to train neural nets to assign 651 accessions to ecotypes. Neural networks have high minimum data requirements. Why not just skip the whole complication of *in silico* assignment (since there's not enough data to train anyway) and use visual assignment by experienced switchgrass breeders for the three ecotypes? Or at least have breeders assign ecotypes on a much larger portion of the diversity panel so that there might be enough data to train a neural network?

2. The authors mention phenology (line 607) in the Methods as one of the traits used to discriminate ecotypes. What phenological characteristics were collected on the panel? Does it include flowering time? If not, why not? If so, it is curious why there is no discussion of flowering time at all in this study, given that biomass and climate adaptation were the primary foci of the work. Flowering time is a strong determinant of biomass since biomass accumulation occurs during the vegetative stage of flowering plants, and latitude-associated climatic factors (photoperiod and thermal time) directly drives flowering time. Flowering time cannot be disregarded especially when the study involves collecting diverse accessions and planting them out in common gardens that span such a large latitudinal range.

It would have been interesting to see whether GWAS results of flowering time co-localize with any of the GWAS intervals detected here for climate or biomass. The GWA could be done either directly from data in the current study (if flowering time was collected) or using previous GWA studies on switchgrass flowering time if those data were not collected in the present study (see below). In any case, flowering time cannot be ignored when discussing climate adaptation across latitudes; it behooves the authors to consider phenology.

REFERENCE

Grabowski PP, Evans J, Daum C, Deshpande S, Barry KW, Kennedy M, Ramstein G, Kaeppeler SM, Buell CR, Jiang Y, Casler MD. Genome-wide associations with flowering time in switchgrass using exome-capture sequencing data. *New Phytologist*. 2017 Jan;213(1):154-69.

3. Extended data figure 2. One of the main conclusions of the biomass and survival analyses is that both biomass and survival of the diversity panel in northern sites is closely associated with where these genotypes originated. Can the authors provide some explanation as to the marked difference between FRMI (site ID IL) and KBSM (site ID MI) (values 67.3% and 95.2%, respectively, in panel a) when they are the most closely related sites climatically to each other (panel b) and they are both northern sites?

4. Figure 3a. I'd be interested in seeing a map that displays the actual origin of the geo-referenced

samples in addition to (or in place of) the maps that show the simulated ranges from SDM.

5. Figure 2c. No discussion why 'mildest 25%' outperform 'coldest 25%' even at northern-most latitudes in the upland ecotype (blue)?

Minor comments

1. Supplemental data 6. Would be more helpful to present values of dry biomass in kg rather than already log-transformed values and use the site ID (e.g., TX1, MI etc) rather than the four letter code for the site. Additionally, it would facilitate re-use of these data if presented in long vertical format (e.g. biomass and survival as two columns with site ID as a third).

2. Extended data figure 3c. Can actual r^2 values be shown in addition to predicted? Yes, there will be a lot of points but maybe they can be made gray/translucent with the predicted r^2 as a solid line on top.

3. Figure 2d. It would be helpful to label the gardens (at least on the biomass axis) with their two letter site IDs.

4. Figure 3b. Since this is a main figure it would be more useful to have the actual descriptor of the climatic variables rather than 'bio16', 'bio2' etc..

Author Rebuttals to Initial Comments:

REFEREE #1:

In the manuscript entitled "Polyploidy and genomic introgressions facilitate climate adaptation and biomass yield in switchgrass" Lovell et al., present an impressive dataset comprised of a high-quality switchgrass genome, and a large set of ecotypes (732) that they re-sequenced and phenotyped across 10 common garden sites across the USA. The tetraploid genome is chromosome scale and resolved into sub-genomes, which the authors leverage to generate a SNP diversity map across three-types of tetraploid ecotypes. The authors leverage machine learning to categorize the ecotypes into phenotypic classes to deconvolute genetic background, site of collection and growth response across the common garden experiment. The authors develop a GWAS methodology to handle the sheer volume of data generated and demonstrate that switchgrass has extensive and differential loci controlling biomass and climate of origin fitness. Next the authors looked specifically at putative introgressions and found that Midwest contributed to the Atlantic subpopulations and that this accounted for climatic adaption of the former population. Finally, the authors present an interesting paradox where the K sub-genome shows dominance, yet the N sub-genome shows more biomass SNP heritability and introgressions.

The authors expertly present a phenomenal amount of complex data with both attractive and informative figures. The writing is clear and concise, although at times could be a bit more descriptive in both the main text and methods. For instance, many tools are cited and only sometimes explained what they do; while some reviewers/reader may be familiar with some of these tools, they most likely are not familiar with all.

- **We fully agree. We have significantly expanded the methods to more completely describe various tools employed here. This includes our comparative genomes pipeline [446-449], gene annotation methods [432-437], ecotype assignments (see below) [594-633], genomic introgressions [645-653], and population genetics [575-579]. We have also expanded our field methods instead of referencing previous work from our group [359-367].**

Also impressive is the rigor of the analyses, which are well designed with accompanying comprehensive statistical support. One criticism is that the title is basically a truism—most people already agree that polyploidy and introgression facilitate climate adaptation—hence the term ecotype.

- **It is true that many people assume that polyploidy and introgression facilitate climate adaptation; however, as reviewer #2 & 3 point out, our work here is one of the most in-depth experimental verifications of these views presented to date.**
- **We have explored more generic titles, but have decided to stick with a title that states the primary discoveries in the study.**

There are many recent reviews in covering both topics that despite the extensive reference list the authors don't include to provide context to their finding.

- **We have expanded the introductions/interpretations of results and the discussion. This is particularly significant for discussion of the adaptive nature of introgressions [245-246, 254-257] and polyploidy [267-268, 280-286]. See our response below to reviewer 3's suggestion for more interpretation regarding population genetics.**

What the authors do bring to light is the extent of the introgressions and polyploid paradox in a mostly undomesticated yet potentially economically important bioenergy “crop.” For instance, what makes this study more than an enormous amount of data and a high-quality genome that support what most breeders/geneticists/ecologists already believe (although I agree that believing and proving are quite different)? To that end, one of the potentially missed opportunities or confounding features is that the authors only look at SNPs. While the panicum genomes (and most grass genomes) are highly colinear, questions of polyploidy (fractionation), sub-genome dominance, and introgression all beg the question of the variation that is not in the reference genome. While the methods are state of the art or better in terms of the variation analysis, the authors do have an incredible amount of Illumina sequencing (median=59x) that could be used for *de novo* assembly and gene content analysis (despite the complexity of the switchgrass genome).

- **Thanks for this comment. As you suggest, these reference-free analyses are doable with the level of coverage we have in our resequencing data. We have completed analysis of both presence-absence variation (PAV) and structural variants (SV) [129-131, 171-173].**
- **However, we are hesitant to rely too heavily on these results because *de novo* (reference free) SV and especially PAV detection can be potentially unreliable in polyploids and outbred genomes — the homeologs/homologs may be similar enough to indicate presence of genes when in fact you are only seeing the alternative haplotype of the other subgenome homeolog. Nonetheless, we were able to produce high-quality SV and PAV calls for 252 libraries.**
- **Despite the limitations of reference-free variant calls in outbred polyploids, we agree that there is an opportunity to use these resources. To this end, we have added a database of high-confidence PAVs and SVs that are proximate to significant GWAS peaks or within introgression intervals.**
- **We have also included a population genetic analysis that leverages and compares the three types of variants. The population structure is similar and we present this result in a new panel in Extended Data Fig. 3.**

Also, this reviewer was left hanging with the excitement as to what genes/loci/variation were underlying the GWAS and introgression peaks. Understandably it would be highly speculative, and some may say descriptive, although since the loci are differential across the GWAS peaks there may be commonalities that are counterintuitive much like the K/N sub-genome dominance paradox.

Lines 201-206: Are the usual suspects found under the loci for climate and fitness. Having a high-quality reference and gene prediction (and RNAseq) provide an opportunity to confirm known or even better yet identify loci previously not associated with these traits. Since there was minimal overlap across the identified

regions it would be interesting if the loci are from similar gene ontology categories or expression network modules; ie different genes but similar pathways.

Lines 218-222: What are the locations of the overlap between the GWAS and introgressions? Seems like this should be a very interesting gene/pathway list.

[addressing these three comments together here]

- We agree that candidate gene discussion is speculative, especially from such a large list of regions: there are >10k GWAS peaks and 4,114 introgression intervals, each with multiple potential causal loci. This number of regions is really too large to make meaningful biological inference from enrichment tests. While in principal, we do not take issue with candidate gene exploration, we feel that we cannot do potential causal locus exploration justice in the main text of this manuscript.
- We feel that what is valuable about the candidate genes is not the cursory discussion that we could present in the main text, but instead the database of genes. These could supply a lifetime of targets for molecular characterization and follow up. To this end, we have developed a complete set of candidate genes, which now includes several additional lines of evidence that permits ranking of candidate genes. This dataset is included in the supplementary material and accompanying methods have been added [772-790]. The additional candidate gene information we now present includes:
 1. Physical proximity to significant GWAS peak / introgression midpoint.
 2. Putative effect of SNPs on each gene within each genetic subpopulation: since GWAS and introgression analyses were conducted within genetic subpopulations, we expect candidates to have functional variants within the subpopulation in which they were discovered.
 3. Minor allele frequencies of presence-absence and structural variants in each genetic subpopulation (see below).
 4. Your point regarding RNAseq and other analyses that could rank candidate genes is well taken. We now include gene coexpression subgraph assignments and GO terms. Co-evolving genes with similar expression patterns may offer a set of more likely genes.
 5. Co-localization of GWAS and introgression candidates
- We now provide a short summary [208-210] of this candidate gene exploration in the main text.

Line 95: “..large, repetitive, polyploid and heterozygous..” What distinguishes switchgrass in this sentence? Wheat is much larger and more repetitive, and higher ploidy. Would be good to put this comment in context because 1.1 Gb (or 2.2 Gb) is not that large of a genome. Also, the level of heterozygosity is not mentioned.

Line 107-108: The use of the wording “for the first time,” really requires something special. A haplotype and sub-genome resolved tetraploid would merit a statement like this. How does this assembly compare to the two closely related tetraploid broomcorn millet genomes (*Panicum miliaceum*)?

Line 97: define “complete” and how did you establish this?

Extended Data Figure 1a what is the basis of the claim that this genome is among the best available plant genomes? Would be great, especially in the extended data, to give the reader some type of gauge for how difficult a plant genome and the accomplishments with the switchgrass genome compared to what is currently out there. The authors have done such a great job with providing the community with high quality genomes in the past, this is a great opportunity to highlight it with more than a throw away sentence in a legend.

[addressing these four comments together here]

- We are of two minds about comparisons of our AP13 genome assembly quality to other published genomes. While it is very true that there are larger or more repetitive (e.g. wheat) or higher ploidy (e.g. strawberry) genomes out there that are nearly as high quality, AP13 v5 really is the first outbred ‘platinum’ genome with all of these complexities.
- As suggested, to make some comparisons, we have added a table to extended data figure 1, which compares the genome assembly quality of a selection of polyploid or outbred genomes.

- The closest similarly complex genome with higher assembly quality is the new *Populus trichocarpa* genome; however, the poplar genome is roughly 1/3rd the size of switchgrass.
- The only truly outbred polyploid genome we include is strawberry. Despite its larger genome size, the switchgrass contig N50 is ~70x longer than the new strawberry genome.
- Also, our contig N50 is ~15x higher than the inbred *P. miliaceum*, which, since it is inbred, has an effective genome size <1/2 of switchgrass.
- However, we don't want to detract from the main points of this manuscript relating to evolution and breeding by emphasizing these genome comparisons in the manuscript. To this end, we have dialed back the interpretation of how novel and excellent our AP13 reference genome is. Some specific changes:
 1. We add that genome sequencing efforts in outbreeding species typically use inbred accessions as their references (e.g. Maize B73) [92-94]. Such inbreeding would not be representative of the genetic diversity in switchgrass, and we now provide a supplementary figure showing that AP13 is within the normal range (albeit on the lower end) of observed heterozygosity in our diversity panel.
 2. All this said, we have also toned-down comparisons to other genomes throughout, including dropping clauses like 'for the first time', 'best available' and 'complete', which are potentially overstatements.
 3. We provide statistics for the heterozygosity and %repetitive sequence for AP13 [93-94].
 4. We state that we do have a subgenome-resolved haploid assembly [108-109].

Line 98: N50=5.5M base pairs, awkward wording-maybe just spell it all out: "megabases." Wasn't the N50 initially 1.1 Mb in the methods (line 439)? Also, is it N50 or L50? Different between the two places.

- The discrepancy between the two N50 results has to do with the primary (more contiguous) and alternative (more fragmented) paths through an outbred assembly. The 1.1Mb value was for all contigs, regardless if they made it into the primary or alternative haplotype. The 5.5Mb is for the primary. We now discuss this in the methods [388-390, 428-429].
- Good catch on L50/N50. For clarity, we have removed any mention of L50.

Line 125-126: why not de novo assemble and anchor to the contigs to look at the later mentioned introgressed regions? Why not do a pan genome analysis with this great number of ecotypes? Are there regions that are unique to the resequencing that cannot be accessed other than by assembly?

Lines 233-236: What was the relative gene retention in the introgression regions? This is where de novo assembling the ecotypes and anchoring them to the introgressed regions may reveal interesting present/absence variation.

[addressing these two comments together here]

- This is a clever idea and one we have now explored [530-552, 787]. We have generated de novo assemblies for a subset of resequenced libraries with high and consistent coverage and extracted presence absence (PAV) and structural variants (SV).
- While these data do offer a unique perspective on genome-wide patterns of evolution (Extended data fig. 3), exploration of specific regions requires extreme caution. This is because, even in the simplest case of an inbred diploid species, short read de novo assemblies are typically riddled with gaps. Gaps may contain false-positive gene absences and bias local inference of gene PAV and SV — the more gaps and smaller the contigs, the more false-absences. While this could be controlled for *in silico* in some systems, we do not believe that this is appropriate in switchgrass. This is because, in outbred genomes, the level of heterozygosity is a strong predictor of contiguity. To split haplotypes, each overlapping pair of reads must share ≥ 1 heterozygous variant. In homozygous stretches, the contigs collapse, producing a gap in one haplotype. So, PAV in any given interval may be a product of heterozygosity as much as true presences/absences of genes. Since heterozygosity can easily be impacted by introgressions, we do not feel that it is appropriate to discuss an over or under-abundance of gene retention in the introgression regions.
- These caveats aside, we did explore both PAV and SV genome-wide and in the introgression and GWAS regions (see below), and we include these data in the new candidate gene supplementary

table (see candidate gene response above). Overall, there is some signal and it looks like this may be a fertile line of inquiry when we have long-read *de novo* genomes across switchgrass diversity.

Unless I missed it (which is possible since there was a lot of information and inaccessible content) the phenotypic data was not described. The predictive methods were explained but not how the measurements were taken etc. Or is the second paragraph in the first section the “phenotyping” section? If so, where is the description of how leaf phenotypes were taken (lines 131-135).

- **The description/method for ecotype characterization from phenotypes was lacking. Reviewer #2 & 3 also brought this up. See our response to reviewer #2’s comments below. In short, we have now significantly expanded the methods describing the ecotype assignment analysis, phenotyping methods and source data [592-631]. We have also added a supplementary table with the ecotype classification model specifications.**

Lines 263-265: Is it possible that sub-genome K might have variation not detected by SNP/resequencing analysis?

- **A subgenome-specific SNP calling bias is a possibility we considered. Prior to the subgenome-heritability bias analysis, we explored resequencing coverage in each library for each pair of homeologous chromosomes. We now show that there is no difference in coverage between the N and K subgenome across libraries [127-128, 526-529]. We believe this to be strong evidence that there should not be a variant calling bias between subgenomes.**

Lines 269-284: The discussion could place the finding in the context of more recent reviews/results across wild plants and crops (as mentioned above). It would be interesting get more insight into the significance of having a less dominant genome having the introgressions and important variation.

- **We fully agree. We have expanded the discussion section [296-299] and discussion/contextualization of results throughout.**

Figure 4c, why are the dots in the ocean?

- **There are lines connecting to their georeferenced locations (so that the points don’t overlap). We have increased the line thickness to make this clear.**

The extended data tables were not openable (looked like binary or some other format). Downloaded several times to check. Could not review the content.

- **Sorry that the compression made the SI inaccessible. The new supplementary files are uncompressed plain text to improve accessibility.**

REFEREE #2:

(Remarks to the Author): I found this manuscript to be outstanding for the following reasons. It uses novel methods to present novel insights on topics of broad interest and societal importance. Specifically, the authors coupled multiple lines of evidence (genomic markers, plant morphology and performance, weather indicators) to show not only that switchgrass is doing surprising things with implications for our understanding of plant evolution, but that plant breeders might be going about improving plants in the wrong way to rapidly develop crops in a changing climate. The methods with which I am most familiar in this paper, the common garden and weather modeling work, are top-notch and frankly boggle the mind. Pulling off a study of this magnitude is herculean but is exactly what was needed to reveal holistic patterns in plant form, function, genetics, survival, and heritability that have been limiting perennial grass improvement. This study avoids the limitations of many other studies by measuring the things that matter in the locations that matter; it isn't a snapshot of performance in a given location, nor is it a meta-analysis with confounding variables. One of the coolest things about this study is the insight that non-dominant sub-genomes may be a repository for survival genes, and that a benefit of polyploidy is the relaxed regulation of the sub-genome that allows genes that may not confer fitness at the moment to persist until conditions arise in which they do confer fitness. I took away from this study that the sub-genome can act like your attic or your basement: you rarely clean it, but there are things there that might be needed someday. I found the discussion in this paper to be supported by the evidence presented and written to clearly advance science. Particularly salient was the authors' observation that science's current tendency to avoid complex systems like obligate outcrossing, polyploidy species in favor of simpler model systems might be misguided for plant improvement in a changing climate.

Before publication I would like to see details of the morphometric scales the team used to measure plant morphology and ecotypes across genotypes and common gardens. To my knowledge, there is no agreed-upon, systemic scale available to the scientific community to measure switchgrass features consistently.

- **This is a very good point and one that reviewers #1 and 3 also brought up. We fully agree that a systematic approach that can identify ecotypes from readily phenotyped traits would be a boon to the switchgrass community.**
- **To generate robust and reproducible ecotype classifications, we opted for an integration between subjective expert/breeder assignments and *in silico* classification from multi-site phenotype data. By using phenotypes collected at gardens near both the northern and southern range margins, we sought to minimize the GxE that could obscure more generalizable ecotype classes.**

This work is not repeatable if we don't know how they measured their plants.

- **Upon revisiting the ecotype classification section of our methods, it is clear that we did not provide enough details. We have now considerably expanded this section with detailed descriptions about how traits were measured [594-604]. We also provide a detailed representation of how the subjective breeders' upland-lowland classification was conducted [605-613]. Full descriptions of formulas and more detail on measurements can be found in the new supplementary data table.**

Further, it was not clear how morphometric data was used to determine ecotype differences. I would like to see a supplement that gives repeatable instructions for how morphology was measured, and how metrics led to categorization as upland, lowland, or coastal. I believe such a supplement will greatly increase the citation rate for this paper by providing a scale and methods the rest of the perennial grass community can use.

- **We have significantly expanded the methodological description of the classification models [614-632].**

- **To aid in the repeatability of this classification model, we have included an additional supplementary table that contains the neural network parameters and the variable weightings in the linear discriminant analysis.**
- **We have also added a supplementary figure showing in more detail how the *in silico* ecotype classification meshes with subjective breeder's ecotype scales.**

Referee #3 (Remarks to the Author):

In this work, the authors present a version 5 assembly and annotation of the switchgrass genome, 'AP13', which is characterized by several complexities including its large size, high degree of heterozygosity, and polyploid and repetitive nature. One marked improvement in this v5 assembly and annotation was the collinearity between the N and K sub-genomes. Methodology presented here would be relevant for other complex genomes, most especially for other polyploid plant species. The authors go on to investigate the genetic basis of climate adaptation, by assembling and re-sequencing a diversity panel of 732 tetraploid switchgrass genotypes, which were phenotyped for survival and biomass production, among other traits (morphology, phenology, etc) in 10 common gardens that span 1862km of latitude. Ecotypes were assigned to the majority of the panel, and these assignments were also included as variables for subsequent analyses. The main conclusion from the survival and biomass analyses was that accessions tend to perform best in climates that are most similar to where they originated, and that climate-of-origin variables related to temperature, especially 30-year minimum temperature, were the best predictors of survival in the common garden trials. Population structure of the diversity panel was next investigated and compared to ecotype assignments; the primary finding here is that ecotypes and subpopulations were not as aligned as previously assumed. The authors also perform climate and biomass GWAS to understand the contribution of loci from each sub-genome to climate adaptation and further investigated whether the prevalence of upland ecotypes in both MIDWEST and ATLANTIC genetic subpopulations was due to independent origins or admixture with a substantial set of analyses. They discovered that MIDWEST introgressions into northern ATLANTIC accessions were more likely to contain significant GWAS intervals shared between the two subpopulations, supporting the hypothesis that introgression from MIDWEST facilitated northern expansion of the ATLANTIC subpopulation.

Overall, the manuscript is well-written and the presentation logical. It is clear that a substantial amount of work went into the study. The new assembly and annotation of AP13 along with the re-sequenced diversity panel is sure to provide the switchgrass community with a valuable new genetic resource whose benefits will continue to be realized for a long time to come. With respect to the evolutionary and ecology questions addressed, the finding that accessions tend to perform best under conditions most similar to the climate from where they originate is not novel. However, the careful dissection of the potential contribution of introgressions to facilitate subpopulation range expansion was well done. The weakest part of the study, in my opinion, is related to ecotype assignment and the general treatment of trait data. These are outlined below.

Specific Comments

Concerns related to ecotype assignment. In the Methods, it states that only seven cultivars were used to train neural nets to assign 651 accessions to ecotypes. Neural networks have high minimum data requirements.

- **Good catch. Our methods in this section were incomplete. Seven total training genotypes would indeed be an unacceptably low number to train a set of 651. What we neglected to state, but have now included [620-626], is that the seven representative genotypes were used to find a set of 85 genotypes that were most closely aligned to the cultivars with known ecotypes. We believe that a training set of 85 is sufficiently large for our purposes here.**

Why not just skip the whole complication of *in silico* assignment (since there's not enough data to train anyway) and use visual assignment by experienced switchgrass breeders for the three ecotypes? Or at least

have breeders assign ecotypes on a much larger portion of the diversity panel so that there might be enough data to train a neural network?

- At first, this is exactly what we did. We now include a new extended data figure showing experts' ecotype classifications against the *in silico* classification.
- However, we soon learned that phenotypes in different gardens vary non-linearly among genotypes. Such genotype-by-environment interactions are not easily captured by breeder's ecotype scores. However, these patterns are easily distinguished in multivariate classification methods.
- To integrate these two approaches, we built our neural network on the 32 garden-specific phenotypes as well as two breeder-defined ordinal variables that distinguish the ecotypes (see response to reviewer #2 above). These are much more completely detailed in the methods and specifically highlighted in the extended data figures.
- We believe that this *in silico* integration between objective phenotypic assays and subjective breeders' assignment offers the best balance between forces outlined in your comment.
- Despite our preference for the neural network assignment, we have also included a *de novo* classification model built on k-means clustering and discriminant analysis of principal components (DAPC). These classifications have been added to the ecotype assignment supplementary table and methods have been added accordingly [614-620].

The authors mention phenology (line 607) in the Methods as one of the traits used to discriminate ecotypes. What phenological characteristics were collected on the panel? Does it include flowering time?

- We used two measures of phenology (now detailed in the methods [597-598] and the new SI table): date of green-up (when the first green tissue emerges from the winter-dormant rhizome crown) and date of panicle emergence (when the first reproductive structures emerge from the tiller). Panicle emergence is highly correlated with flowering time, but serves as a more consistent proxy since the date of floral opening is influenced strongly by short duration heatwaves or droughts, which adds environmental variance to measures of flowering time.
- These traits used for ecotype assignment were not used for GWAS or other analyses.

If not, why not? If so, it is curious why there is no discussion of flowering time at all in this study, given that biomass and climate adaptation were the primary foci of the work. Flowering time is a strong determinant of biomass since biomass accumulation occurs during the vegetative stage of flowering plants, and latitude-associated climatic factors (photoperiod and thermal time) directly drives flowering time. Flowering time cannot be disregarded especially when the study involves collecting diverse accessions and planting them out in common gardens that span such a large latitudinal range.

- We initially considered integrating biomass and phenology in this analysis (as we did previously with genetic mapping in <https://www.pnas.org/content/pnas/116/26/12933.full.pdf>). However, as we found in that analysis, the link between phenology and biomass is not linear and is mediated both by genetic covariance with other traits (e.g. winter survival, duration of flowering, etc.) and complex genotype-by-environment interactions.
- We have found that phenology certainly can be a major correlate of biomass and fitness. However, biomass and fitness are not only a consequence of phenology, but also a host of other traits. While in some sites flowering time may indeed be the primary phenotypic driver of biomass, this is not necessarily the case in most of our gardens. For example, winter survival is by far the largest driver of fitness/biomass in our northwestern sites. We also have data showing that pathogen resistance is a major driver of biomass in southern sites.
- In the end, we agree that flowering date is an essential component of switchgrass improvement and plant fitness. Though we did, in fact, measure flowering date in 2019, the genetics of flowering as a response to environmental cues varies in a complex fashion among genetic subpopulations across these common gardens. Since this is a paper about biomass and fitness, we feel that singling out phenology as the only non-biomass/survival trait would be inappropriate. A manuscript with these results is in preparation and is the next priority for submission.

It would have been interesting to see whether GWAS results of flowering time co-localize with any of the GWAS intervals detected here for climate or biomass. The GWA could be done either directly from data in the current study (if flowering time was collected) or using previous GWA studies on switchgrass flowering time if those data were not collected in the present study (see below). In any case, flowering time cannot be ignored when discussing climate adaptation across latitudes; it behooves the authors to consider phenology.

REFERENCE: Grabowski PP, Evans J, Daum C, Deshpande S, Barry KW, Kennedy M, Ramstein G, Kaepler SM, Buell CR, Jiang Y, Casler MD. Genome-wide associations with flowering time in switchgrass using exome-capture sequencing data. *New Phytologist*. 2017 Jan;213(1):154-69.

- **As stated above, we believe that including GWAS on phenology is beyond the scope of this paper.**
- **However, we do agree that looking at overlaps with other published GWAS analyses could prove to be an interesting line of inquiry. Dr. Grabowski, who is a coauthor on this manuscript, has extracted the proximate genes to peaks in the 2017 NP paper, which used the version v1.1 switchgrass assembly and annotation. Since v1.1 is not subgenome-specific, we cannot determine which homeologs of a candidate gene in v1.1 corresponds to the correct v5 homeolog. Furthermore, v1.1 is not contiguous enough for synteny-constrained searches, so we are left with 1-copy(v1):2-copy(v5) orthology networks of note. Nonetheless, we did parse these and found a number of potentially interesting candidates, which have been flagged in our newly added candidate gene lists. However, we feel that the incongruence between the two reference genomes and the nature of orthology to an unphased polyploid genome does not permit strong enough inference to warrant candidate gene discussion in the main text, even with the necessary hedging.**

Extended data figure 2. One of the main conclusions of the biomass and survival analyses is that both biomass and survival of the diversity panel in northern sites is closely associated with where these genotypes originated. Can the authors provide some explanation as to the marked difference between FRMI (site ID IL) and KBSM (site ID MI) (values 67.3% and 95.2%, respectively, in panel a) when they are the most closely related sites climatically to each other (panel b) and they are both northern sites?

- **This is a good point and one that we should have addressed more clearly. We have now added a panel to extended data figure 2 showing the severity and duration of cold snaps during the winter of 2018-19. We hope it is obvious that the west-east gradient of severe cold in 2018-19 does not necessarily match generic climate similarity, which includes other seasons which would not affect winter survival. We have added a sentence in the main text to this effect [145-146] and discuss the daily temperature data acquisition and analysis in the methods [673-678].**
- **Additionally, the KBSM site receives much more snow (lake effect) than the other sites, which could have insulated the rhizomes from an otherwise killing cold. However, we do not have the snowpack depth data that could support this claim, and felt that it was inappropriate to include this circumstantial evidence in the text.**

Figure 3a. I'd be interested in seeing a map that displays the actual origin of the geo-referenced samples in addition to (or in place of) the maps that show the simulated ranges from SDM.

- **Fig. 1A has points showing the georeferenced localities of collection for all genotypes; however, this was not clear in the caption, nor were the points particularly obvious in the plot. We have adjusted both the caption and plot contrast, hopefully resolving this confusion.**

Figure 2c. No discussion why 'mildest 25%' outperform 'coldest 25%' even at northern-most latitudes in the upland ecotype (blue)?

- **Happy you asked this. We omitted a discussion of this pattern. This omission has been resolved [153-157]. In short, (nearly) all uplands survived everywhere in 2019 and upland plants from the south appear to generally do better than those in the north. It is possible that a more intensely cold winter than 2018 could introduce differential survival in the uplands and produce a tradeoff similar to what we observed in the two more southern genotypes.**

Minor comments

Supplemental data 6. Would be more helpful to present values of dry biomass in kg rather than already log-transformed values and use the site ID (e.g., TX1, MI etc) rather than the four letter code for the site.

Additionally, it would facilitate re-use of these data if presented in long vertical format (e.g. biomass and survival as two columns with site ID as a third).

- **Agreed. We wanted to be sure to present log-transformed so that the analyses could be easily replicated. We now include both raw and transformed data in the SI.**

Extended data figure 3c. Can actual r^2 values be shown in addition to predicted? Yes, there will be a lot of points but maybe they can be made gray/translucent with the predicted r^2 as a solid line on top.

- **The raw data is far too large to plot. There are 75M points that went into this analysis. Really no matter how much transparency we add, it looks like a complete cloud. Maybe as a happy medium, would it suffice to use a much narrower interval in the plots (2bp instead of 500)? This does produce a somewhat viewable cloud of points. We have added these smaller window results to the plot.**

Figure 2d. It would be helpful to label the gardens (at least on the biomass axis) with their two letter site IDs.

- **This plot is not straightforward; the axes are not specific gardens.**
- **The garden ID depends on the genotype. For example, the right most column (most climatically similar garden) might be MI for a plant collected in Michigan, but TX3 for a genotype collected in Louisiana. We tried to clarify this using the annotations/arrows and have expanded the figure caption.**

Figure 3b. Since this is a main figure it would be more useful to have the actual descriptor of the climatic variables rather than 'bio16', 'bio2' etc..

- **This is a bit tricky, since it requires quite a bit of description (which we provide in the ext data fig). We tried to add descriptions into the axis and couldn't do it without making the figure too messy to really understand. In lieu of descriptions on the axis, we have added a sentence in the caption to see the ext. data fig with the full descriptions. Hopefully this is satisfactory.**

Reviewer Reports on the First Revision:

Referee #1 (Remarks to the Author):

The authors have addressed all of my comments and concerns. Thank you for adding the additional methods and analyses.

Referee #2 (Remarks to the Author):

The authors have addressed my concerns regarding repeatability of the plant morphology measurements.

Emily Heaton

Referee #3 (Remarks to the Author):

In this work, the authors present a revision to their initial submission on the assembly and

annotation of the switchgrass genome, AP13, and subsequent investigation on the genetic basis of climate adaptation through assembling, re-sequencing and phenotyping a diversity panel of 732 tetraploid switchgrass genotypes. The authors have submitted satisfactory clarifications and modifications in response to my most of my previous comments and in reading the revision, I was reminded of the impressive amount of work that went into this research. This is all commendable.

I thank the authors for providing the updated methods on how they assembled their training dataset for ecotype classification. I like the new inclusion of the experts' classification results; are the colors in Extended Data Figure 2e the classification based on NN (as seems to be suggested in the response to reviewers document that the extended figure shows "experts' ecotype classifications against the in silico classification")? If that is the case, please indicate this in the caption so that readers may understand that this coloring cross-references the other analysis.

My one lingering suggestion, perhaps more for the Editor rather than the authors, is to have the ecotype classification methodology reviewed by an expert on neural networks to ensure that the training data is indeed sufficient. Re-iterating my previous comment, NNs have very large data requirements for training, without which, the models will be over-fit. While 85 is improved from the original 7 observations reported, the method of selecting those additional 78, i.e. selecting other observations that are most similar to the original seven (based on genetic groupings, phenotypic groupings and state-of-origin [L624-626]), I am not sure gives rise to a more functionally expanded training set, despite the increase in the total number. In any case, it would be helpful to have these types of details be reviewed by someone who is an expert in NN.

Referee #4 (Remarks to the Author):

I enjoyed reading the manuscript and appreciate the substantial scope of the study. My review focused specifically on the ecotype classification methodology.

The description of phenotypic measurements are now clear and sufficiently detailed to be reproduced.

The addition of the PCA and qualitative scores provide corroboration of the neural-network classification method. Cross-validation of the neural network would be stronger, but I appreciate that this is difficult given the small datasets size.

Minor comments:

85 samples is still a small training set for a neural network and therefore overfitting is a potential concern. It may be helpful to mention that the neural network itself is very small (1 hidden layer with 5 units) within the methods text, because using a low capacity neural network helps to reduce the potential for overfitting.

The methods text should clarify that the neural network was trained on 34 features: 16 quantitative traits at 2 locations, as well as the 2 qualitative traits.

It may also be helpful to clarify that the 2 qualitative traits used as input to the neural network are distinct from the qualitative expert score 1-5 which is used to corroborate the neural network predictions in Extended Fig. 2e.

[line 628] Please clarify that "caret" was used to implement the neural network.

[line 799] In addition to the code for the GWAS analyses, please make the code for DAPC and the neural network analyses available.

Minor typo:

Supplemental table "Variable calculation for neural net":
"upland leaf/tiller traits" TIL+LEA (PKLE) -> LEAF

Author Rebuttals to First Revision:

REFEREES' COMMENTS:

1. Referee #1 (Remarks to the Author): The authors have addressed all of my comments and concerns. Thank you for adding the addition methods and analyses.
2. Referee #2 (Remarks to the Author): The authors have addressed my concerns regarding repeatability of the plant morphology measurements. Emily Heaton
3. Referee #3 (Remarks to the Author): In this work, the authors present a revision to their initial submission on the assembly and annotation of the switchgrass genome, AP13, and subsequent investigation on the genetic basis of climate adaptation through assembling, re-sequencing and phenotyping a diversity panel of 732 tetraploid switchgrass genotypes. The authors have submitted satisfactory clarifications and modifications in response to my most of my previous comments and in reading the revision, I was reminded of the impressive amount of work that went into this research. This is all commendable. I thank the authors for providing the updated methods on how they assembled their training dataset for ecotype classification.
 - a. I like the new inclusion of the experts' classification results; are the colors in Extended Data Figure 2e the classification based on NN (as seems to be suggested in the response to reviewers document that the extended figure shows "experts' ecotype classifications against the in silico classification")? If that is the case, please indicate this in the caption so that readers may understand that this coloring cross-references the other analysis.
 - i. **We have now included a statement to this effect in the figure caption.**
 - b. My one lingering suggestion, perhaps more for the Editor rather than the authors, is to have the ecotype classification methodology reviewed by an expert on neural networks to ensure that the training data is indeed sufficient. Re-iterating my previous comment, NNs have very large data requirements for training, without which, the models will be over-fit. While 85 is improved from the original 7 observations reported, the method of selecting those additional 78, i.e. selecting other observations that are most similar to the original seven (based on genetic groupings, phenotypic groupings and state-of-origin [L624-626]), I am not sure gives rise to a more functionally expanded training set, despite the increase in the total number. In any case, it would be helpful to have these types of details be reviewed by someone who is an expert in NN.
 - i. **Thank you for your thoughtful comments. See response to reviewer #4 below.**
4. Referee #4 (Remarks to the Author): I enjoyed reading the manuscript and appreciate the substantial scope of the study. My review focused specifically on the ecotype classification methodology. The description of phenotypic measurements are now clear and sufficiently detailed to be reproduced. The addition of the PCA and qualitative scores provide corroboration of the neural-network classification method. Cross-validation of the neural network would be stronger, but I appreciate that this is difficult given the small datasets size.
 - a. Minor comments: 85 samples is still a small training set for a neural network and therefore overfitting is a potential concern. It may be helpful to mention that the neural network itself is very small (1 hidden layer with 5 units) within the methods text, because using a low capacity neural network helps to reduce the potential for overfitting.
 - i. **We have added a statement to this effect in the methods.**
 - b. The methods text should clarify that the neural network was trained on 34 features: 16 quantitative traits at 2 locations, as well as the 2 qualitative traits. It may also be helpful to clarify that the 2 qualitative traits used as input to the neural network are distinct from the qualitative expert score 1-5 which is used to corroborate the neural network predictions in Extended Fig. 2e.
 - i. **Added to the methods**
 - c. [line 628] Please clarify that "caret" was used to implement the neural network.
 - i. **Added to the methods**

- d. [line 799] In addition to the code for the GWAS analyses, please make the code for DAPC and the neural network analyses available.
 - i. **We have added a new github link the code availability section that includes these scripts.**
- e. Minor typo: Supplemental table “Variable calculation for neural net”: "upland leaf/tiller traits" TIL+LEA (PKLE) -> LEAF
 - i. **Fixed**